# Long noncoding RNA *AGPG* regulates PFKFB3-mediated tumor glycolytic reprogramming

Jia Liu[1,6], Ze-Xian Liu [1,6], Qi-Nian Wu[1,6], Yun-Xin Lu[1,6], Chau-Wei Wong[2,6], Lei Miao[1], Yun Wang[1], Zixian Wang[1], Ying Jin[1], Ming-Ming He[1], Chao Ren[1], De-Shen Wang[1], Dong-Liang Chen[1], Heng-Ying Pu[1], Lin Feng [1], Bo Li [3], Dan Xie [1], Mu-Sheng Zeng [1], Peng Huang[1], Aifu Lin[4], Dongxin Lin[1], Rui-Hua Xu [1,5✉] & Huai-Qiang Ju [1,5✉]

Tumor cells often reprogram their metabolism for rapid proliferation. The roles of long noncoding RNAs (lncRNAs) in metabolism remodeling and the underlying mechanisms remain elusive. Through screening, we found that the lncRNA Actin Gamma 1 Pseudogene (*AGPG*) is required for increased glycolysis activity and cell proliferation in esophageal squamous cell carcinoma (ESCC). Mechanistically, *AGPG* binds to and stabilizes 6-phosphofructo-2-kinase/fructose-2,6-biphosphatase 3 (PFKFB3). By preventing APC/C-mediated ubiquitination, *AGPG* protects PFKFB3 from proteasomal degradation, leading to the accumulation of PFKFB3 in cancer cells, which subsequently activates glycolytic flux and promotes cell cycle progression. *AGPG* is also a transcriptional target of p53; loss or mutation of *TP53* triggers the marked upregulation of *AGPG*. Notably, inhibiting *AGPG* dramatically impaired tumor growth in patient-derived xenograft (PDX) models. Clinically, *AGPG* is highly expressed in many cancers, and high *AGPG* expression levels are correlated with poor prognosis, suggesting that *AGPG* is a potential biomarker and cancer therapeutic target.

[1] State Key Laboratory of Oncology in South China, Collaborative Innovation Center for Cancer Medicine, Sun Yat-sen University Cancer Center, Guangzhou 510060, China. [2] The First Affiliated Hospital of Sun Yat-sen University, Guangzhou 510080, China. [3] Department of Biochemistry and Molecular Biology, Zhongshan School of Medicine, Sun Yat-Sen University, Guangzhou 510080, China. [4] College of Life Sciences, Zhejiang University, Hangzhou 310058, China. [5] Precision Diagnosis and Treatment for Gastrointestinal Cancer, Chinese Academy of Medical Sciences, Guangzhou 510060, China. [6]These authors contributed equally: Jia Liu, Ze-Xian Liu, Qi-Nian Wu, Yun-Xin Lu, Chau-Wei Wong. ✉email: xurh@sysucc.org.cn; juhq@sysucc.org.cn

Rapid proliferation and glucose metabolism remodeling are hallmarks of cancer. To provide sufficient energy and support rapid biosynthesis, cancer cells exhibit enhanced glycolysis, even under normoxic conditions; this phenomenon is referred to as the Warburg effect[1]. During glycolysis, glycolytic intermediates can be diverted to the biosynthesis of macromolecules, including nucleotides, amino acids, and fatty acids, which are necessary for cancer cell proliferation and tumor progression. Characterizing the cooperative mechanisms underlying glycolysis and cell proliferation could lead to a better understanding of human cancer development. Long noncoding RNAs (lncRNAs) are suggested to be involved in metabolic reprogramming, but the mechanisms remain elusive[2,3]. Our recent studies investigated the roles of metabolic reprogramming in promoting glycolysis and redox hemostasis[4–7]. The roles of lncRNAs in metabolism remodeling and the underlying mechanisms have thus attracted our interest.

Phosphofructo-2-kinase/fructose-2,6-biphosphatase 3 (PFKFB3) catalyzes the production of fructose-2,6-bisphosphate (F-2,6-BP), a potent allosteric stimulator of the key enzyme 6-phosphofructokinase 1. Thus, the activation of PFKFB3 has been linked to enhanced glycolysis[8]. Interestingly, PFKFB3 is mainly localized in the nucleus[9], which is different from other members of the PFKFB family. Recent studies revealed the unexpected role of PFKFB3 in promoting cell proliferation by regulating the expression of important cell cycle proteins: cyclin-dependent kinase-1 (CDK1) is upregulated, and p27 is downregulated, partially owing to the nuclear delivery of F-2,6-BP[10]. As reported, PFKFB3 inhibition is a promising modality for cancer treatment because it suppresses glycolysis, proliferation, and metastasis in cancer cells[11–13].

In recent years, many lncRNAs have been identified to regulate cancer metabolism, but the underlying mechanisms remain elusive. Here, we identified that the lncRNA **A**ctin **G**amma 1 **P**seudo**g**ene (*AGPG*) has a pivotal role in glucose metabolism remodeling and cell proliferation by enhancing PFKFB3 stability. Intriguingly, this is the first lncRNA shown to directly bind to and regulate PFKFB3. The *AGPG*-PFKFB3 interaction protects PFKFB3 from ubiquitination and proteasomal degradation, thus promoting glycolysis and cell cycle progression at the G1/S phase transition. We also demonstrated that p53 binds to the *AGPG* promoter and represses its transcription, indicating that *AGPG* is a target of p53. Moreover, high *AGPG* expression levels are correlated with poor overall survival in esophageal squamous cell carcinoma (ESCC), suggesting that *AGPG* may be a biomarker and therapeutic target for ESCC treatment.

## Results

### Identification of *AGPG* as a metabolism-related lncRNA.
To find oncogenic lncRNAs that significantly affect ESCC development, we first identified lncRNAs that were more highly expressed in ESCC tissues than in paired adjacent normal tissues from The Cancer Genome Atlas (TCGA) database. Then, we sorted these lncRNAs according to the log2-fold change. Next, we built an siRNA library targeting the top 50 lncRNAs (Supplementary Fig. 1a). For the siRNA screening, the siRNA library was designed with the SMARTselection algorithm to ensure high-efficiency silencing. These siRNAs also contained the proprietary ON-TARGETplus dual-strand chemical modification to ensure optimal strand loading and to disrupt microRNA-like seed activity, thereby reducing off-target effects. To pinpoint lncRNAs that might alter glucose metabolism, we transfected the siRNA library into two human ESCC cells and examined cell viability and lactate production. We found 14 lncRNAs that might be required for cell proliferation, 10 involved in lactate

production and 8 potentially involved in both cell viability and glucose metabolism (Fig. 1a). Among these eight lncRNAs, *AGPG* knockdown significantly decreased cell viability and lactate production (Fig. 1b, Supplementary Fig. 1b). Bioinformatics analysis revealed that *AGPG* is located on chromosome 1q32.1 and has 3 exons (1–56, 10,447–10,526, and 11,304-13,488) (Supplementary Fig. 1c). We focused on the isoform AC098934.2-201, and for simplicity, we refer to this isoform as *AGPG*. According to the coding potential calculator, the coding potential of *AGPG* is very low.

Then, we verified *AGPG* expression levels in a panel of human ESCC cells and normal esophageal epithelial cells (Het-1A and NE-1). We found that *AGPG* levels were significantly higher in the tumor cells than in the normal cells, and the copy number of *AGPG* was also increased in ESCC cells (Fig. 1c, d, Supplementary Fig. 1d). The functional role of *AGPG* in cell proliferation and lactate production was further verified in other ESCC cell lines (Supplementary Fig. 1e–g).

### *AGPG* expression correlates with prognosis of ESCC.
Consistent with our bioinformatics analysis results (Fig. 1e), we found that high *AGPG* levels were correlated with an unfavorable overall survival in ESCC patients in an independent cohort (Fig. 1f; Sun Yat-sen University Cancer Center (SYSUCC), $n = 122$; clinicopathological information is provided in Supplementary Table 1). We categorized gene expression as low or high in comparison with the median value: if the expression level was higher than the median, it was classified as high, whereas if it was lower than the median, it was low. Multivariate analysis also indicated that *AGPG* was an independent prognostic factor in ESCC patients (Supplementary Table 2).

As suggested by the TCGA database analysis, *AGPG* was highly expressed in multiple types of cancer, including gastric cancer (GC), colorectal cancer (CRC), liver cancer, breast cancer, and lung cancer (Supplementary Fig. 1h). However, in some cancers, such as glioblastoma multiforme (GBM), head and neck squamous cell carcinoma and thyroid carcinoma, there was no significant difference in expression between tumor and normal tissues. In addition, *AGPG* levels were decreased in cancers such as kidney chromophobe, kidney renal clear cell carcinoma, and kidney renal papillary cell carcinoma. Similar to many other lncRNAs, *AGPG* has tissue-specific expression patterns in different cancers[14]. Next, we performed qRT-PCR and RNAScope in situ hybridization (ISH) assays to detect *AGPG* expression in ESCC, GC, and CRC tissues[15]. Our results showed that *AGPG* was highly expressed in ESCC, GC, and CRC tissues (Fig. 1g–i). These results suggest a strong relationship between *AGPG* dysregulation and cancer development.

To identify the subcellular localization of *AGPG*, we detected *AGPG* expression in cytoplasmic and nuclear fractions by qRT-PCR analysis. The results showed that *AGPG* was localized predominantly in the nucleus, with some localization in the cytoplasm, which was further verified by RNAScope ISH and RNA fluorescence in situ hybridization assays[16] (Fig. 1j–l, Supplementary Fig. 1i).

### *AGPG* promotes cell proliferation and glycolysis.
Because *AGPG* is potentially involved in cell proliferation and lactate production, we further investigated the functional role of *AGPG* in cellular behaviors. *AGPG* knockdown in KYSE150 and KYSE30 cells strikingly inhibited cell proliferation and colony formation (Fig. 2a–d, Supplementary Fig. 2a, b), and *AGPG* knockdown blocked the G1/S cell cycle transition (Fig. 2e, f, Supplementary Fig. 2c). We also detected key cell cycle proteins and observed that *AGPG* knockdown markedly increased p27 expression and

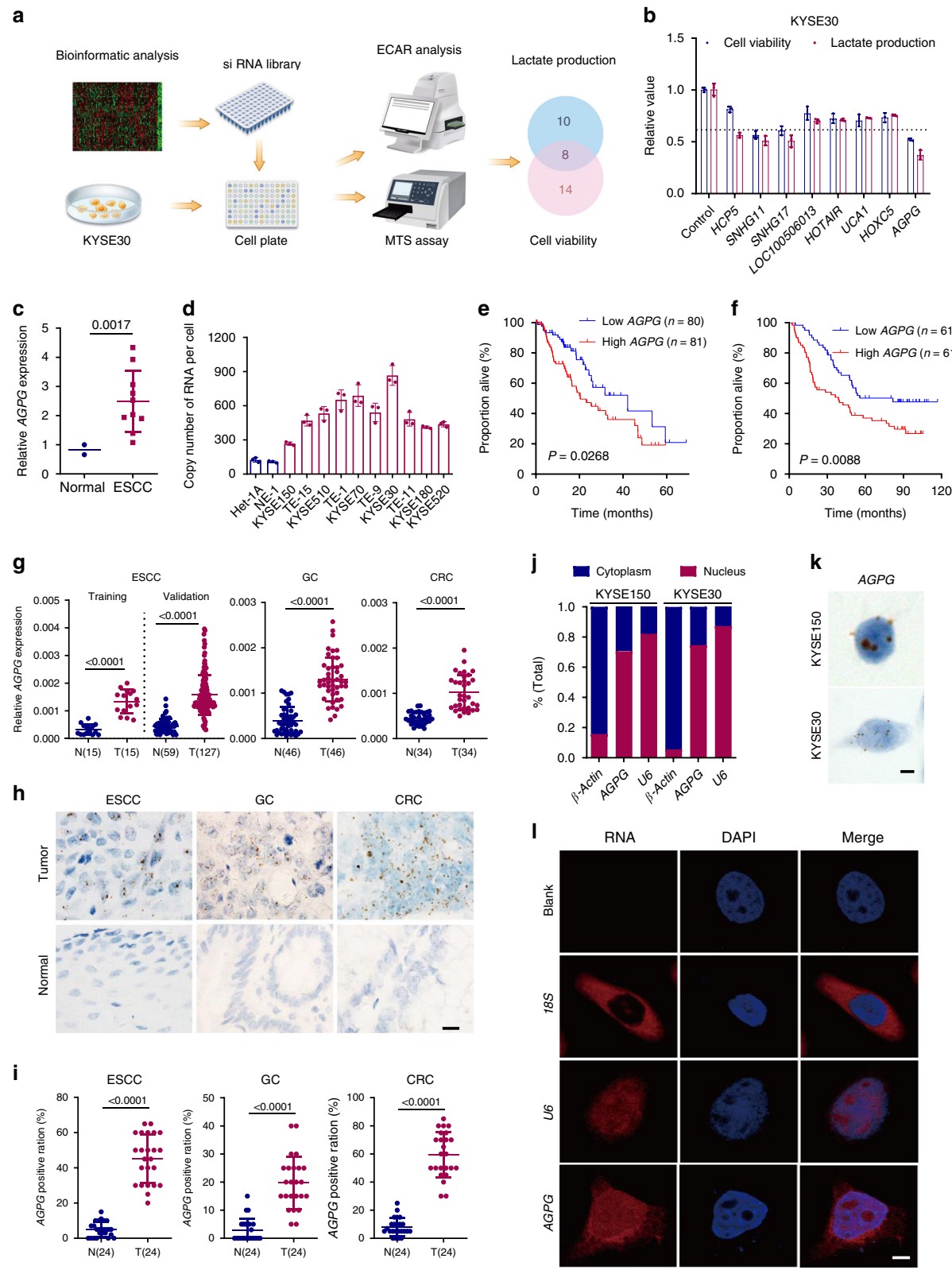

decreased CDK1 expression (Fig. 2g) but did not change the expression of p21, p53, CDK3, or CDK6 (Supplementary Fig. 2d). These results suggest that *AGPG* might regulate key cell cycle proteins, including p27, and CDK1-mediated G1/S progression to promote proliferation.

To verify the role of *AGPG* in metabolic reprogramming, the extracellular acidification rate (ECAR) of ESCC cells was measured using a Seahorse XF24e Extracellular Flux Analyzer (Fig. 2h, i). We demonstrated that *AGPG* knockdown significantly impaired glycolysis, which was consistent with our previous screening results. To further determine the metabolic flux of glucose, we detected intracellular amounts of $^{13}C$-labeled metabolic intermediates in ESCC cells after incubation with $^{13}C_6$-glucose for 2 h (Fig. 2j). Metabolome

**Fig. 1 Identification of *AGPG* as a metabolism-related lncRNA. a** Experimental scheme for identifying lncRNAs potentially involved in both cell viability and glucose metabolism. **b** Eight lncRNAs regulated both cell proliferation and lactate production in KYSE30 cells, *n* = 3 biologically independent samples. **c** qPCR detection of *AGPG* expression in multiple ESCC cells (*n* = 10 cells) and in normal esophageal epithelial cell lines (*n* = 2 cells). **d** Determination of *AGPG* copy number, *n* = 3 biologically independent samples. **e** Overall survival analysis based on *AGPG* levels in ESCC (TCGA, *n* = 161, log-rank test, two-sided). **f** Overall survival analysis based on *AGPG* levels in ESCC detected by qPCR (SYSUCC, *n* = 122, log-rank test, two-sided). **g** qPCR detection of *AGPG* expression in ESCC (training set *n* = 15, validation set *n* = 59, 127, respectively), GC (*n* = 46), CRC (*n* = 34) and normal tissues. **h** RNAScope ISH detection of *AGPG* expression in ESCC, GC, CRC, and matched normal tissues. Scale bar: 20 μm. **i** RNAScope ISH detection and statistical analysis of AGPG expression in ESCC, GC, CRC, and matched normal tissues. Data are presented as mean±S.D., *n* = 24 cases per tissue type, the *p* value was determined by a two-tailed unpaired Student's *t* test. **j** qPCR detection of *AGPG* expression in the cytoplasmic and nuclear fractions. **k** RNAScope ISH detection of *AGPG* subcellular localization. Scale bar: 5 μm. **l** Subcellular localization of *AGPG* detected by FISH. Scale bar: 5 μm. Data in **b**–**d**, **g**, **i** are representative of three independent experiments and presented as mean±S.D., the *P* value was determined by a two-tailed unpaired Student's *t* test.

analysis based on liquid chromatography and mass spectrometry (MS) showed that intracellular metabolites of glycolysis (3-phosphoglycerate, pyruvate, and lactate) were markedly decreased after *AGPG* knockdown, further confirming that *AGPG* is essential for the conversion of glucose to lactate[17] (Fig. 2k–m).

To further confirm the functional role of *AGPG*, we generated *AGPG* CRISPR KO cells using the CRISPR/Cas9 genome-editing system (Supplementary Fig. 2e). Consistently, *AGPG* CRISPR KO significantly inhibited ESCC cell proliferation and cell cycle progression (Supplementary Fig. 2f, g). In addition, *AGPG* CRISPR KO led to a significant reduction in aerobic glycolysis (Supplementary Fig. 2h). Taken together, our data indicated that *AGPG* is functionally important in regulating cancer metabolic reprogramming and tumor growth.

**AGPG is directly associated with PFKFB3**. To dissect the molecular mechanisms underlying *AGPG*-mediated metabolic remodeling, we tried to identify *AGPG*-associated proteins through RNA pull-down assays followed by mass spectrometry. The mass spectrometry data are provided in Supplementary Data 1, 2. We compared the *AGPG*-binding proteins with antisense *AGPG*-binding proteins. Proteins that bound to antisense *AGPG* were excluded from the candidate list, and the remaining proteins were sorted by MS score, as described in previous lncRNA studies[15]. We found that sense *AGPG*, but not the antisense control, interacted specifically with PFKFB3 (Fig. 3a), which was also localized mainly in the nucleus, as previously reported. This observation was further confirmed by the finding that *AGPG* bound directly to purified His-tagged recombinant PFKFB3 (Fig. 3b). The interaction between *AGPG* and PFKFB3 was also confirmed by RNA immunoprecipitation (RIP) assays (Fig. 3c, Supplementary Fig. 3a, b). To further characterize the interaction between *AGPG* and PFKFB3 in vivo, we performed MS2-tagged RNA affinity purification (MTRAP) and western blotting. Compared with expression of the negative control, coexpression of the MS2-*AGPG* and MCP-3FLAG plasmids led to significant enrichment of PFKFB3, demonstrating that PFKFB3 specifically binds to *AGPG* (Fig. 3d).

Immunofluorescence colocalization analysis showed that *AGPG* and PFKFB3 colocalized mainly in the nucleus, with some colocalization in the cytoplasm, which suggests that the *AGPG*–PFKFB3 complex may play roles in both the nucleus and cytoplasm (Fig. 3e). Moreover, we examined *AGPG* expression by qPCR and PFKFB3 expression by western blotting in a panel of ESCC cells and 12 pairs of ESCC tissues and matched normal esophageal tissues (SYSUCC). As expected, *AGPG* expression was positively correlated with PFKFB3 expression in ESCC (Fig. 3f, Supplementary Fig. 3c), which further implied the functional relationship between *AGPG* and PFKFB3. In addition, PFKFB3 was more highly expressed in ESCC tissues than in normal tissues, as previously reported.

Absolute quantification of *AGPG* and PFKFB3 levels showed that there were ~ 400–700 *AGPG* molecules per cell versus ~ 4400–7400 PFKFB3 molecules per cell (Supplementary Fig. 3d), indicating that there are sufficient *AGPG* copies in ESCC cells[18]. Collectively, these results suggest that *AGPG* and PFKFB3 are closely related and that their interaction plays an important role in human cancer development.

**The T5 fragment of *AGPG* mediates the interaction with PFKFB3**. Computational secondary structure analysis revealed that *AGPG* contains five main branches (Supplementary Fig. 3e). To map the regions that mediate the interaction of *AGPG* with PFKFB3, we performed RNA pull-down assays using in vitro-synthesized full-length (FL) *AGPG* and T1 (1–800), T2 (801–1140), T3 (1141–1700), T4 (1701–2030), and T5 (2031–2321) fragments and then analyzed the products by western blotting. We demonstrated that the T5 fragment could bind to PFKFB3, whereas the other fragments or the beads-only control could not. These results were further verified with purified recombinant PFKFB3 (Fig. 3g). We also performed crosslinking IP and qPCR (CLIP-qPCR), an improved method for the isolation of lncRNA segments bound by PFKFB3; consistently, the T5 fragment was identified as the main region responsible for binding PFKFB3 (Fig. 3h, Supplementary Fig. 3f).

After deleting the T5 fragment, *AGPG* could no longer interact with PFKFB3 (Fig. 3i). Overexpression of *AGPG* FL, but not a mutant lacking the T5 fragment (*AGPG* ΔT5), was sufficient to prevent the phenotypes observed after *AGPG* knockout (KO), including glycolytic reprogramming and cell proliferation (Fig. 3j, k, Supplementary Fig. 3g, h). These results suggest that the T5 fragment is required for *AGPG* to interact with and regulate PFKFB3, and the downstream cellular processes are probably mediated through the interaction of *AGPG* with PFKFB3.

To further identify the specific motif that are responsible for PFKFB3 binding, we performed crosslinking-IP and high-throughput sequencing (HITS-CLIP). Hypergeometric Optimization of Motif EnRichment (HomeR) was used for motif analysis based on the binding peaks obtained by Piranha and CIMS analyses[19,20]. The RNA motifs recognized by PFKFB3 are listed in Supplementary Table 3. Among these motifs, CCAGCCA or similar motifs were highly ranked and could be identified by multiple methods (Fig. 3l). Compared with the input, PFKFB3 CLIP enriched more reads mapping to the *AGPG* sequence around the identified motif. To further verify whether this motif coordinates PFKFB3 binding, we performed RNA pull-down assays using wild-type *AGPG* (WT) and CCAGCCA motif-deleted *AGPG* (MT). We demonstrated that the binding of a mutant lacking the CCAGCCA motif (*AGPG* MT) and PFKFB3 was significantly reduced (Supplementary Fig. 3i). In addition, overexpression of this *AGPG* MT could not rescue the decreased glycolysis caused by *AGPG* KO (Supplementary Fig. 3j). These data suggest that the CCAGCCA motif of *AGPG*

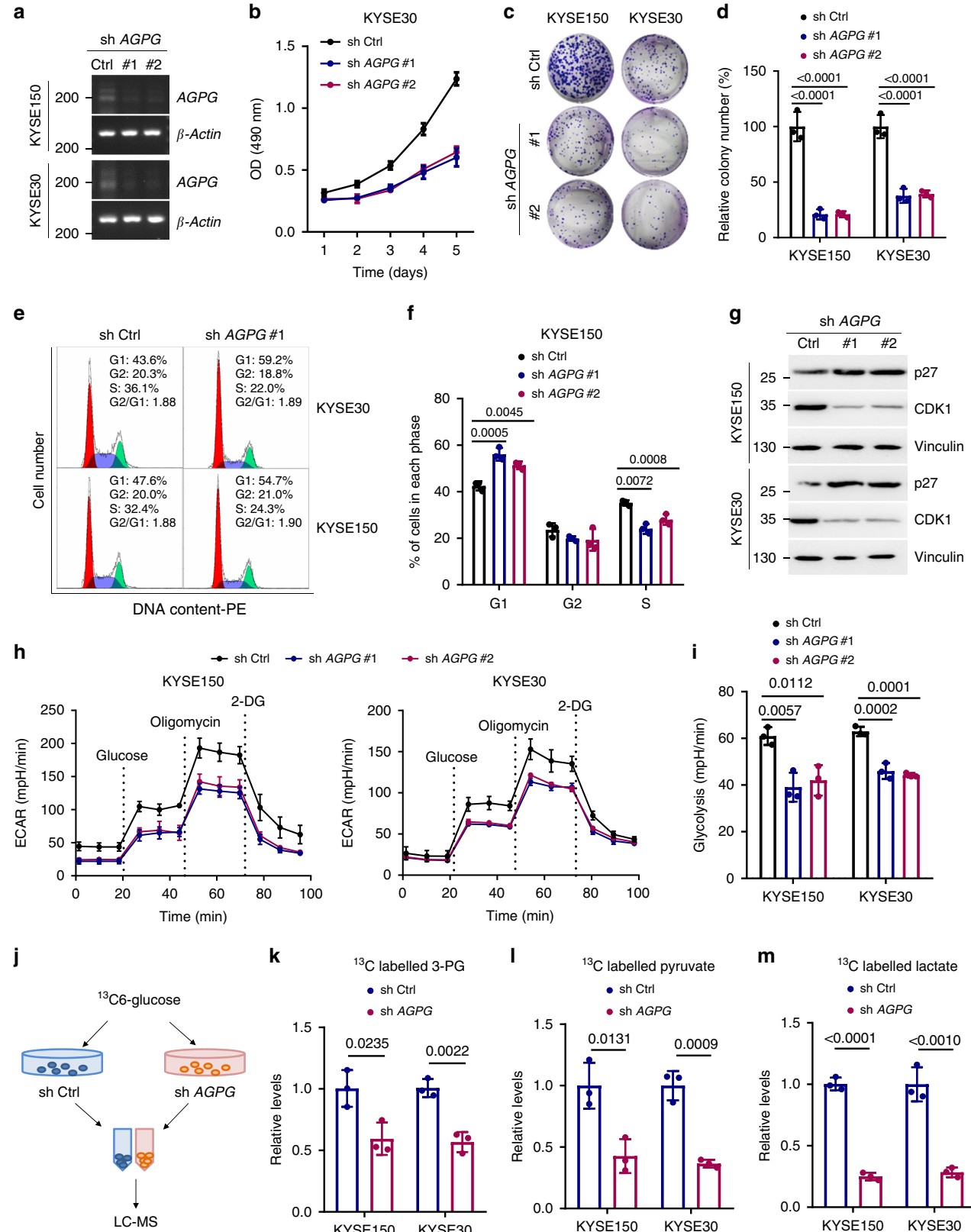

is important for its ability to bind PFKFB3 and promote tumor glycolytic reprogramming.

### *AGPG* blocks APC/C-mediated PFKFB3 ubiquitination.

Because PFKFB3 contains a kinase domain and a phosphatase domain, three human FLAG-tagged PFKFB3 vectors harboring FL (1–520), N-terminal (N, 1–245) and C-terminal (C, 246–520) constructs were constructed to identify the PFKFB3 residues that associated with *AGPG*. Interestingly, we demonstrated that *AGPG* interacted mainly with the C-terminal fragment, with minimal binding with the N-terminal fragment (Fig. 4a, Supplementary Fig. 4a). Then, we performed RIP assays using an anti-FLAG

**Fig. 2 AGPG is required for cell proliferation and metabolism remodeling. a** qPCR and electrophoresis detection of *AGPG* expression in KYSE30 and KYSE150 cells. Ctrl, control. **b** Cell proliferation was assessed by MTS assays (OD 490 nm). **c, d** Colony formation assays and statistical analysis of ESCC cells transduced with sh*AGPG* #1 or #2 or shCtrl. **e** The cell cycle was analyzed by flow cytometry analysis. **f** Statistical analysis of KYSE150 cells (%) in each cell cycle phase. **g** CDK1 and p27 expression levels were detected by western blotting in cells transfected with sh*AGPG* #1 or #2 or shCtrl. **h** The ECAR was measured in cells transfected with sh*AGPG* #1 or #2 or shCtrl using an XF Extracellular Flux Analyzer. **i** Statistical analysis of the effects of *AGPG* knockdown on glycolytic activity. **j** Flowchart of the experiments for identifying the role of *AGPG* in glucose metabolism. **k–m** $^{13}$C-Labeled metabolic intermediates of glycolysis were decreased after *AGPG* knockdown. Data in **b, d, f, i, k–m** are representative of three independent experiments and presented as mean±S.D., $n = 3$ biologically independent samples, the $P$ value in **b, d, f, i** was determined by one-way analysis of variance (ANOVA) with Dunnett's multiple comparisons test, no adjustments were made for multiple comparisons. The $P$ value in **k–m** was determined by a two-tailed unpaired Student's $t$ test.

antibody or IgG control. As expected, *AGPG* precipitated with the C-terminal fragment of PFKFB3 (Fig. 4b).

Then, we evaluated the functional effect of the *AGPG*-PFKFB3 interaction on PFKFB3. Interestingly, *AGPG* knockdown mediated by shRNA significantly reduced the expression of PFKFB3 (Fig. 4c), which was also confirmed in *AGPG* CRISPR KO cells using CRISPR/Cas9 (Fig. 4d). In addition, overexpression of *AGPG* FL but not *AGPG* ΔT5 rescued the decreased PFKFB3 level induced by *AGPG* deletion (Fig. 4d). These data suggest that *AGPG* may regulate the PFKFB3 protein levels through the T5 fragment.

Previous studies have revealed that PFKFB3 is subjected to constant proteosomal degradation through polyubiquitination[21]. Therefore, we hypothesized that the observed effects might be attributable to proteasomal degradation, as decreased PFKFB3 levels were recovered by the proteasomal inhibitor MG-132 (Fig. 4e). In addition, *AGPG* knockdown had marked effects on PFKFB3 stabilization in ESCC cells, shortening the half-life of PFKFB3 (Fig. 4f, Supplementary Fig. 4b, c). Moreover, we carried out IP assays in cells expressing FLAG-tagged PFKFB3 with an anti-FLAG antibody and detected ubiquitin levels by western blotting. As shown, *AGPG* knockdown significantly increased the levels of ubiquitinated PFKFB3 (Fig. 4g, Supplementary Fig. 4d). These data suggest that *AGPG* is required for PFKFB3 stabilization. In addition, *AGPG* CRISPR KO did not affect PFKFB3 mRNA levels or subcellular location (Supplementary Fig. 4e, f). Regarding PFKFB3 enzymatic activity, *AGPG* CRISPR KO had a mild effect on PFKFB3 S461 phosphorylation (Supplementary Fig. 4g), which is widely reported to be a key regulator of PFKFB3 enzyme activity[22]. Therefore, we speculate that *AGPG* has little effect on PFKFB3 enzyme activity. Taken together, these data indicate that *AGPG*-PFKFB3 binding appears to be important for PFKFB3 protein turnover.

PFKFB3 contains a KEN box that targets proteins for ubiquitylation by the anaphase-promoting complex APC/C[21]. PFKFB3 was shown to be subject to degradation involving APC/C-Cdh1, a cell cycle-regulated E3 ubiquitin ligase. APC/C is composed of multiple subunits; as reported previously, active APC/C could be immunoprecipitated from cells using a monoclonal Cdc27 antibody[21,23]. Therefore, to further elucidate the mechanism by which *AGPG* blocks PFKFB3 ubiquitination, we performed coIP assays using PFKFB3 and Cdc27 antibodies to determine whether *AGPG* affects the interaction between PFKFB3 and active APC/C. *AGPG* CRISPR KO significantly increased the PFKFB3/Cdc27 interaction (Fig. 4h), suggesting that *AGPG* could block the binding of Cdc27 to PFKFB3. Therefore, we speculate that *AGPG* specifically binds to PFKFB3 and blocks its interaction with APC/C, which inhibits APC/C-mediated PFKFB3 ubiquitination and the subsequent degradation.

**AGPG stabilizes PFKFB3 by preventing K302 ubiquitination.** Because *AGPG* binds mainly to the C-terminal fragment of PFKFB3, we analyzed four putative ubiquitinated lysine (K)

residues, namely, K292, K302, K352, and K472, in the C-terminus of PFKFB3 to identify the predominant lysine residue (s) subject to ubiquitination that is (are) affected by *AGPG* (Supplementary Fig. 4h). We mutated these four lysine residues to alanine (A) and performed IP and RNA pull-down assays. Among the four mutants, PFKFB3 K302A showed no increase in ubiquitination in response to *AGPG* knockdown (Fig. 4i, Supplementary Fig. 4i), suggesting that *AGPG* might stabilize PFKFB3 by preventing K302 ubiquitination. Moreover, PFKFB3 K302A, but not other mutants, significantly abrogated APC/C (Cdc27)-induced PFKFB3 ubiquitination in ESCC cells (Supplementary Fig. 4j, k), further indicating that K302 is an important site for APC/C-mediated ubiquitination in PFKFB3. Thus, *AGPG* stabilizes PFKFB3 by preventing APC/C-mediated PFKFB3 K302 ubiquitination.

Furthermore, to ascertain the function of K302 in regulating PFKFB3, FLAG-tagged PFKFB3 wild-type (WT) or K302A was exogenously expressed in ESCC cells; intriguingly, the PFKFB3 K302A mutant showed an extended half-life (Fig. 4j, Supplementary Fig. 4l), indicating that K302 is an important ubiquitination site responsible for PFKFB3 stability. Then, we investigated the role of PFKFB3 in *AGPG*-mediated cell proliferation and glycolytic reprogramming. Consistently, PFKFB3 K302A overexpression could significantly reverse the inhibition of glycolysis, cell proliferation and cell cycle progression by *AGPG* CRISPR KO, whereas PFKFB3 WT could only partially rescue these effects (Fig. 4k–m, Supplementary Fig. 4m, n). Taken together, these data show that by inhibiting PFKFB3 K302 ubiquitination, *AGPG* enhanced PFKFB3 stability and therefore led to the increased accumulation of PFKFB3 in cancer cells, thereby increasing F-2,6-BP synthesis and subsequently promoting cell cycle progression by regulating p27 and CDK1. Simultaneously, the increased F-2,6-BP levels activated glycolytic flux by stimulating PFK-1[8,9]. Furthermore, we tested the effect of *AGPG* on *PFKFB3* KO cell lines. After *PFKFB3* KO in ESCC cells, *AGPG* CRISPR KO had mild effects on aerobic glycolysis and cell proliferation (Supplementary Fig. 5a, b). Collectively, these data suggest that the regulatory roles of *AGPG* in cell proliferation and glycolytic reprogramming are mainly dependent on PFKFB3.

**AGPG is a transcriptional target of p53.** Because *AGPG* is highly expressed in tumors, we tried to determine the mechanism of *AGPG* regulation. Pathway analysis indicated that *AGPG* expression was negatively correlated with p53. (Fig. 5a). An analysis of cells with different *TP53* genotypes showed that HCT-116 cells with *TP53* KO displayed higher *AGPG* levels than control cells (Fig. 5b), indicating an inhibitory effect of p53 on *AGPG* expression. Furthermore, cells with WT *TP53* (KYSE150) expressed much lower *AGPG* levels than those harboring mutant (MT) *TP53* (TE-1 and KYSE30)[24] (Fig. 5c). WT *TP53* overexpression in KYSE150 and HCT-116 cells decreased *AGPG* levels, whereas *TP53* knockdown increased *AGPG* expression, further confirming the role of p53 in regulating *AGPG* expression

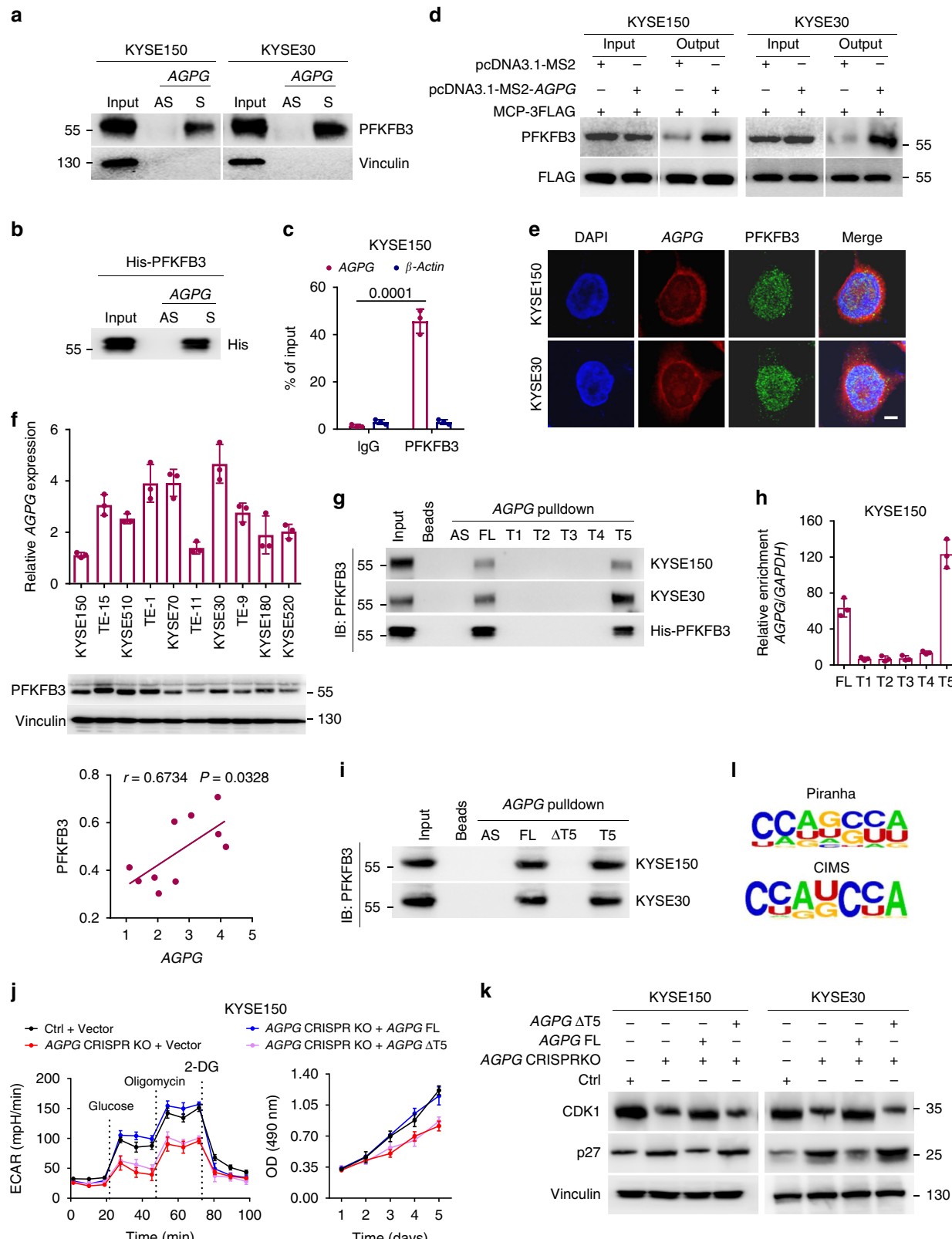

(Fig. 5d). Furthermore, *AGPG* expression level was negatively correlated with *TP53* expression in a cohort of ESCC patients with WT *TP53* by qPCR analysis (Fig. 5e, SYSUCC, n = 72).

Next, we determined the region required for p53-mediated *AGPG* regulation. As expected, the *AGPG* promoter contains a p53-binding sequence (Fig. 5f), which was identified as a

p53-binding region (p53-BR) according to chromatin immuno-precipitation (ChIP) assays[25] (Fig. 5g, Supplementary Fig. 5c). Consistently, the transcriptional activity of luciferase reporters containing an intact p53-BR (p53-BR wt) was markedly weaker than that of those with the p53-BR deleted (p53-BR mt). Moreover, co-transfection of WT *TP53* selectively decreased the

**Fig. 3 *AGPG* directly associates with PFKFB3. a**, **b** PFKFB3 in cell lysates **a** or purified His-tagged recombinant PFKFB3 **b** was pulled down by biotin-labeled *AGPG* but not by *AGPG* antisense RNA. S, sense. AS, antisense. **c** RIP assays indicated that *AGPG* precipitated with PFKFB3 in whole-cell lysates. The RNA levels of *AGPG* and β-actin were measured by qPCR analysis. **d** *AGPG*-binding proteins were detected by MTRAP and western blotting analysis. PFKFB3 bound to *AGPG* was captured by anti-FLAG antibody affinity agarose beads; IP complexes were separated and identified by specific antibodies. **e** Immunofluorescence analysis showed that *AGPG* and PFKFB3 colocalized not only in the nucleus but also in the cytoplasm. Scale bar: 5 μm. **f** qPCR detection of *AGPG* expression and western blotting detection of PFKFB3 expression in human ESCC cells. PFKFB3 expression was positively correlated with *AGPG* expression. (Pearson's correlation analysis, *n* = 10). **g** In vitro-synthesized FL and truncation mutants of *AGPG* were incubated with protein lysates from KYSE150 and KYSE30 cells or with purified His-tagged recombinant PFKFB3. RNA pull-down and western blotting assays were then performed. **h** CLIP-qPCR showed that the T5 fragment of *AGPG* was the region responsible for PFKFB3 binding. **i** RNA pull-down assays showed that *AGPG* ΔT5 could not interact with PFKFB3. **j** *AGPG* CRISPR KO cell lines were generated using the CRISPR/Cas9 genome-editing system. Overexpression of *AGPG* FL, but not of *AGPG* ΔT5, was sufficient to reverse the decreased ECAR and cell proliferation caused by *AGPG* CRISPR KO. **k** Western blotting showed that CDK1 downregulation and p27 upregulation by *AGPG* CRISPR KO were abolished by *AGPG* FL but not by *AGPG* ΔT5. **l** HomeR was used to perform the motif analysis on the binding peaks obtained by the Piranha and CIMS analyses. Both methods suggested that CCAGCCA might be responsible for PFKFB3 binding. Data in **c**, **f**, **h**, **j** are representative of three independent experiments and presented as mean±S.D., *n* = 3 biologically independent samples, the *P* value was determined by a two-tailed unpaired Student's *t* test.

transcriptional activity of reporters with an intact p53-BR (Fig. 5h, Supplementary Fig. 5d). Collectively, our data demonstrate the important regulatory role of p53 in *AGPG* transcription, and loss or mutation of *TP53* leads to the striking upregulation of *AGPG*.

Multiple microenvironmental factors, including hypoxia, DNA damage, and oncogene expression, can affect *TP53* status[26], so we investigated whether these factors are involved in the *AGPG* regulatory network (Fig. 5i). Expression of oncogenic *KRas*[G12V] led to *AGPG* upregulation and *TP53* downregulation in 293 T cells[27] (Fig. 5j), indicating that oncogene stress is involved in *AGPG* regulation by affecting *TP53* status.

We also extended our analysis to hypoxic conditions by detecting *AGPG* and *TP53* expression in cells exposed to severe hypoxia or normoxia for 48 h. Both WT and MT p53 were upregulated as previously reported; *AGPG* expression was decreased in *TP53* WT cells (KYSE150) and was markedly increased in *TP53* MT cells (TE-1 and KYSE30) after exposure to hypoxia (Fig. 5k). These results suggest that hypoxia-induced *AGPG* upregulation could be abolished in the presence of WT *TP53*.

**Effects of *AGPG* on ESCC tumor growth in vivo**. Then, we explored the role of *AGPG* in tumorigenesis in vivo. *AGPG* knockdown significantly repressed cell-based xenograft tumor growth (Fig. 6a–c, Supplementary Fig. 6a, b). As indicated by the Haemotoxylin and Eosin (HE) and immunohistochemistry (IHC) results, *AGPG* knockdown decreased the levels of the cell proliferation marker Ki67, as well as of PFKFB3 and CDK1, but increased p27 levels (Fig. 6d, e, Supplementary Fig. 6c, d); these results are consistent with our in vitro experimental results.

Furthermore, in the patient-derived xenograft (PDX) models (generated using tumor tissues from two ESCC patients, SYSUCC) (Fig. 6f), *AGPG* depletion via in vivo-optimized *AGPG* inhibitor dramatically reduced tumor growth (Fig. 6g–i, Supplementary Fig. 6e–g), implicating *AGPG* as a promising therapeutic target. The main component of the in vivo-optimized *AGPG* inhibitor used in the PDX model was antisense oligonucleotides, which exhibited a stronger knockdown effect on nuclear-localized RNAs[28]. Accordingly, as shown in the HE and IHC analyses, *AGPG* knockdown significantly affected cell proliferation (indicated by Ki67), which might be attributed to the modulation of PFKFB3 and downstream CDK1 and p27 expression, as mentioned previously (Fig. 6j–n).

**The p53-*AGPG*-PFKFB3 axis is involved in ESCC development**. To establish whether the p53-*AGPG*-PFKFB3 axis is clinically associated and pathologically involved in ESCC

development, we detected *AGPG* expression by qPCR and Ki67, PFKFB3, CDK1, p27, and p53 expression by IHC in a cohort of ESCC tissues (SYSUCC, *n* = 102). The *AGPG*-high group exhibited higher Ki67, PFKFB3, and CDK1 expression but lower p27 and p53 expression, whereas the *AGPG*-low group showed the opposite pattern (Fig. 7a, b). Collectively, we speculated that dysregulation of the p53-*AGPG*-PFKFB3 axis promotes ESCC development.

In addition, we examined PFKFB3 expression in a set of ESCC and matched normal tissues by IHC. In agreement with previous reports, PFKFB3 was highly expressed in malignant tissues[29–31] (Fig. 7c, d), and high PFKFB3 expression was associated with poor outcomes in ESCC patients (Fig. 7e, SYSUCC, *n* = 104; clinicopathological information is provided in Supplementary Table 4). We next examined *AGPG* expression in these tissues by RNAScope ISH assays. Then, the tissues were sorted into the *AGPG*/PFKFB3-high, *AGPG*/PFKFB3-intermediate, and *AGPG*/PFKFB3-low groups, among which the *AGPG*/PFKFB3-high subset had a much worse prognosis than the other subsets (Fig. 7f, SYSUCC, *n* = 104; clinicopathological information is provided in Supplementary Table 5). These data further indicate *AGPG*/PFKFB3 as a promising prognostic indicator and a potential therapeutic target.

## Discussion

Cancer cells reprogram glucose metabolism toward aerobic glycolysis to increase their biomass and sustain uncontrolled proliferation[32]. Apart from its well-recognized role in glycolysis regulation, PFKFB3 has been shown in many studies to have an important role in cell cycle progression, in which it catalyzes the production of F-2,6-BP, which functions as a regulator of CDK1 and p27. Thus, PFKFB3 has critical roles in cancer cells by linking glycolysis to cell proliferation. PFKFB3 levels are regulated both transcriptionally and posttranslationally. PFKFB3 is reported to be dimethylated at R131 and R134, and the regulation of PFKFB3 methylation determines directional glucose utilization[17]. The acetylation of PFKFB3 induced by cisplatin impairs PFKFB3 translocation to the nucleus and causes PFKFB3 accumulation in the cytoplasm, leading to increased glycolysis and protecting against DNA damage[33]. Collectively, posttranslational modifications determine the stability and activity of PFKFB3.

At present, the list of lncRNAs involved in tumor progression is rapidly expanding. A few lncRNAs have been implicated in cancer metabolism regulation, but the underlying mechanisms remain poorly understood. Here, we found that the lncRNA *AGPG* significantly influences cell proliferation by directly binding to and stabilizing the key enzyme PFKFB3, which regulates both glucose metabolism and the cell cycle. Canonical RNA-binding proteins

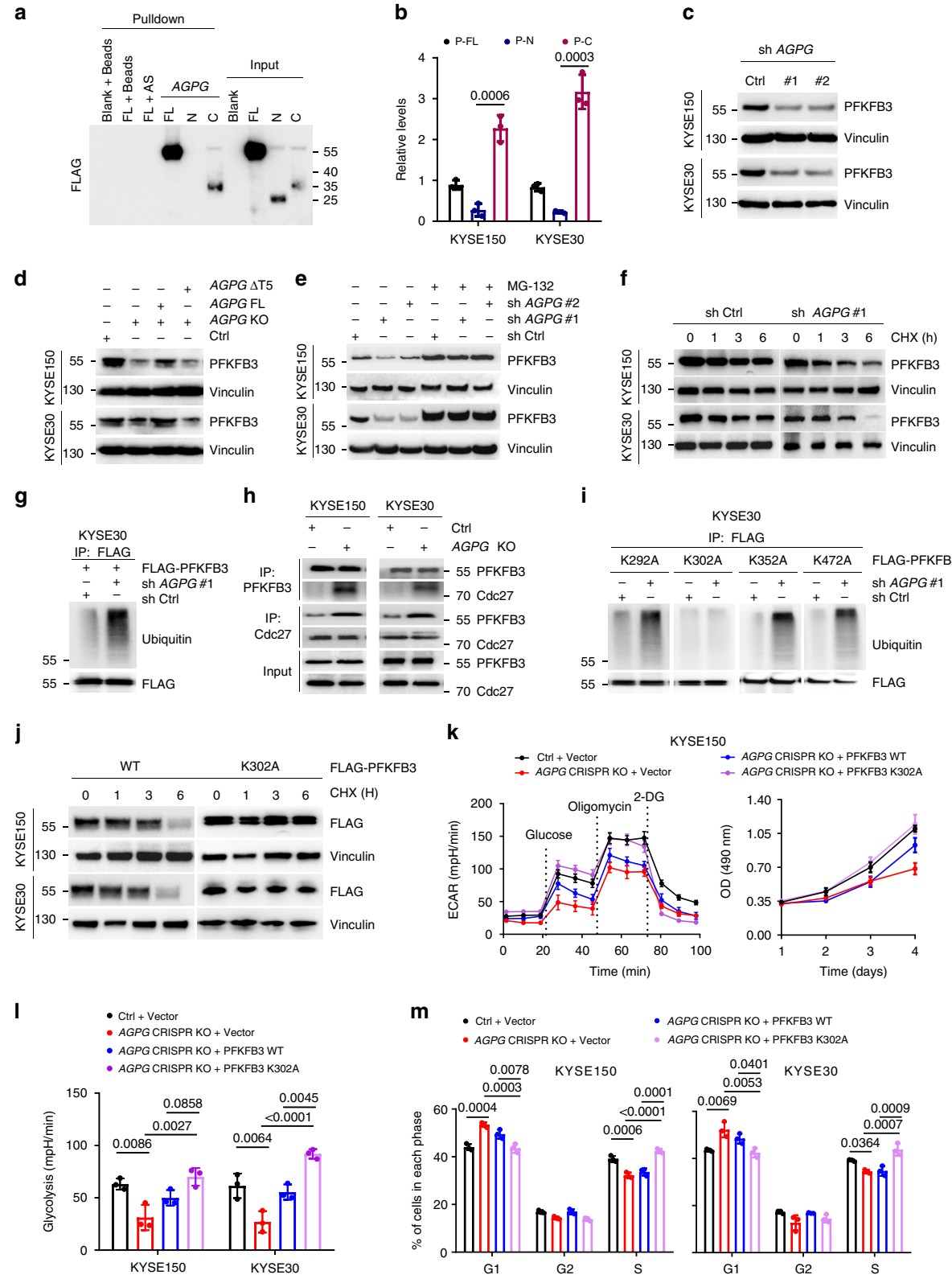

(RBPs) generally contain RNA-binding regions, but recent studies have indicated that hundreds of new RBPs lack known RNA-binding domains, indicating the complexity and diversity of RNA–protein complexes[34,35]. A large number of RBPs are gradually being discovered using various experimental methods. In our study, we performed a variety of in vitro and in vivo experiments to study the interaction between the lncRNA and the

protein, including RNA pull-down, RIP, CLIP, and MTRAP assays. Consistently, we demonstrated that *AGPG* is directly associated with PFKFB3, which implies that PFKFB3 is a newly discovered noncanonical RBP.

The mechanism of PFKFB3 stabilization remains largely unknown. Recent studies have shown that polyubiquitination of PFKFB3 is a critical step leading to degradation through the E3

**Fig. 4 *AGPG* affects PFKFB3 stability by preventing its ubiquitination. a** In vitro-synthesized *AGPG* was incubated with protein lysates from KYSE30 cells transfected with vectors expressing FLAG-tagged FL or truncation mutants of PFKFB3. RNA pull-down and western blotting assays were then performed. Truncation mutants included FL, N-terminal (N) and C-terminal (C) constructs. **b** RIP assays were performed using anti-FLAG antibodies in cells transfected with vectors expressing FLAG-tagged FL or truncation mutants of PFKFB3. **c** *AGPG* knockdown reduced PFKFB3 expression in ESCC cells. **d** PFKFB3 downregulation by *AGPG* CRISPR KO was rescued by *AGPG* FL but not by *AGPG* ΔT5. **e** PFKFB3 downregulation by *AGPG* knockdown was abolished by MG-132 (10 μM, 12 h). **f** Western blotting detection of PFKFB3 levels in KYSE150 cells transfected with shCtrl or sh*AGPG* followed by treatment with CHX (100 μg per ml) for the indicated times. **g** IP assays showed that *AGPG* knockdown increased PFKFB3 ubiquitination levels. FLAG-tagged PFKFB3 was expressed in cells, which were then subjected to IP assays. **h** Active APC/C could be immunoprecipitated from cells using monoclonal Cdc27 antibody. CoIP assays showed that *AGPG* CRISPR KO significantly increased the interaction between PFKFB3 and Cdc27. **i** IP assays showed that *AGPG* knockdown did not increase ubiquitination of the PFKFB3 K302A mutant. **j** Cells were infected with FLAG-tagged PFKFB3 WT or K302A and treated with CHX (100 μg per ml) for the indicated time. FLAG levels were detected by western blotting. **k** PFKFB3 K302A overexpression significantly reversed the decreased ECAR and cell proliferation caused by *AGPG* CRISPR KO, whereas PFKFB3 WT could only partially rescue these effects in KYSE150 cells. **l** PFKFB3 K302A overexpression significantly reversed the decreased glycolysis caused by *AGPG* CRISPR KO, whereas PFKFB3 WT could only partially rescue this effect. **m** PFKFB3 K302A overexpression abolished the G1/S arrest caused by *AGPG* CRISPR KO, whereas PFKFB3 WT could only partially rescue this effect. Data in **b**, **k–m** are representative of three independent experiments and presented as mean±S.D., *n* = 3 biologically independent samples, the *P* value was determined by one-way ANOVA with Tukey's multiple comparisons test. No adjustments were made for multiple comparisons.

ubiquitin ligase APC/C-Cdh1. Our study suggests that *AGPG* specifically binds to PFKFB3 and blocks its interaction with APC/C, which inhibits APC/C-mediated PFKFB3 ubiquitination and subsequent degradation. As reported, PFKFB3 is subject to constant polyubiquitination at several sites, including K142, and proteasomal degradation[21]. Our study also showed that K302 is an important ubiquitination site responsible for PFKFB3 stability, and upon interacting with PFKFB3, *AGPG* blocks K302 ubiquitination and enhances PFKFB3 stability. Collectively, these findings further delineate the detailed mechanism underlying lncRNA-mediated PFKFB3 turnover and cancer metabolism remodeling.

Considering the oncogenic role of *AGPG*, we also investigated the mechanism of *AGPG* regulation, which was associated with the p53 pathway. Cells exposed to severe hypoxia, nutrient deprivation, or genotoxic insults in the tumor microenvironment are characterized by the stabilization and activation of p53[26,36]. Studies have revealed that many lncRNAs are transcriptional targets of p53[25,37,38]. However, the exact p53 target genes responsible for glucose metabolism remain poorly characterized[39,40]. p53 can directly repress transcription by binding p53 response elements[41,42], and this interaction may represent a key regulatory link in the p53-mediated cellular response. In this study, we found that *AGPG* is a transcriptional target negatively regulated by p53 through the identified p53-BR in the *AGPG* promoter. These data further verified the tumor suppressive function of p53 in regulating metabolism. Unfortunately, over 50% of human cancers, including ESCC, have no WT *TP53* function due to mutation or deletion[43,44], and multiple microenvironmental factors, including hypoxia and excessive genotoxic insults, are potential driving forces that cause *TP53* mutations[45]. Therefore, dysregulation of the p53-*AGPG*-PFKFB3 axis leads to metabolism remodeling and cell proliferation.

As a predominant histologic type of malignant esophageal tumor, ESCC is prevalent worldwide, especially in certain regions[46–48]. Although multiple therapies, including surgery, radiotherapy, and chemotherapy, have been used, ESCC remains a leading cause of cancer-related death[49]. Even worse, current studies on targeted therapeutic approaches or biomarker-driven therapies for this malignant disease are not promising. Therefore, it is important to elucidate the molecular mechanisms underlying ESCC and to develop more effective therapies. Our study showed that both *AGPG* and PFKFB3 are highly expressed in ESCC and that high expression of either *AGPG* or PFKFB3 is closely linked to unfavorable outcomes. High expression of both *AGPG* and PFKFB3 is correlated with an even poorer prognosis, suggesting that the combination of *AGPG* and PFKFB3 is a potential

prognostic marker for ESCC diagnosis. In addition to ESCC, *AGPG* is also highly expressed in multiple types of cancer, suggesting that *AGPG* might be a promising pancancer therapeutic target.

In conclusion, our study showed that *AGPG*, a transcriptional target of p53, has a pivotal role in promoting glycolysis and cell proliferation by enhancing PFKFB3 stability, thus facilitating the development of cancer. Our findings provide a basis for RNA interference-based strategies that target lncRNAs and cancer metabolism for cancer treatment.

## Methods

**Cell lines**. Het-1A, NE-1, and HCT-116 cells were obtained from American Type Culture Collection (ATCC, Rockville, MD, USA). KYSE30, KYSE510, KYSE150, KYSE520, KYSE70, and KYSE180 cells were obtained from German Cell Culture Collection (DSMZ, Braunschweig, Germany). TE-1, TE-9, and TE-15 cells were obtained from the Cell Bank of Shanghai Institute of Cell Biology (Chinese Academy of Medical Sciences, Shanghai, China). HCT-116 (*TP53* KO) and 293 T inducible KRas^{G12V} (iK-Ras^{G12V}) cells were provided by professor Peng Huang (SYSUCC, Guangzhou, China). The cells were maintained in RPMI-1640 (HyClone, Logan, UT, USA) supplemented with 10% fetal bovine serum (Invitrogen, Carlsbad, CA, USA) and 1% penicillin/streptomycin (HyClone) at 37 °C with 5% CO$_2$. Based on short tandem repeat (STR) profiling by vendors, no cells used in this study are found in the database of commonly misidentified cell lines. All cells were further authenticated via STR-PCR DNA profiling by Guangzhou Cellcook Biotech Co.,Ltd. (Guangzhou, China) and were determined to be free of mycoplasma contamination.

**Human tissue specimens**. Clinical samples were collected from SYSUCC (Guangzhou, China). All patients had a histological diagnosis of cancer. After the operation, the patients received regular follow-up. All clinicopathological information is provided in Supplementary Tables 1, 4, and 5.

**Reagents, plasmid construction, and site-directed mutagenesis**. MG-132 was purchased from Selleck Chemicals (Houston, TX, USA). Cycloheximide (CHX) and doxycycline were purchased from Sigma-Aldrich (St. Louis, MO, USA). Recombinant human PFKFB3 protein with an N-terminal His tag was purchased from Novus Biologicals (Littleton, Colorado). N-Terminal FLAG-tagged expression vectors (for expression in mammalian cells) for FL PFKFB3, truncated mutants, and site-directed mutants (K292A, K302A, K352A, and K472A) were provided by OBiO Technology (Shanghai, China). Expression vectors for FL *AGPG* and truncated mutants used for in vitro RNA synthesis were provided by OBiO Technology (Shanghai, China). p53 and Cdc27 expression vectors were provided by GeneCopoeia Inc. (Rockville, MD, USA).

**RNA interference**. Cell transfections and lentiviral transductions were performed according to the manufacturer's instructions[7]. Small interfering RNAs (siRNAs) and short hairpin RNAs (shRNAs) were provided by RiboBio (Guangzhou, China) or OBiO Technology (Shanghai, China). The resulting constructs were verified by sequencing. The sequences are listed in Supplementary Table 6.

**CRISPR/Cas9-mediated gene editing**. *AGPG* KO cell lines were generated using a CRISPR/Cas9-based strategy[50]. *AGPG*-specific guide RNA (gRNA) expression

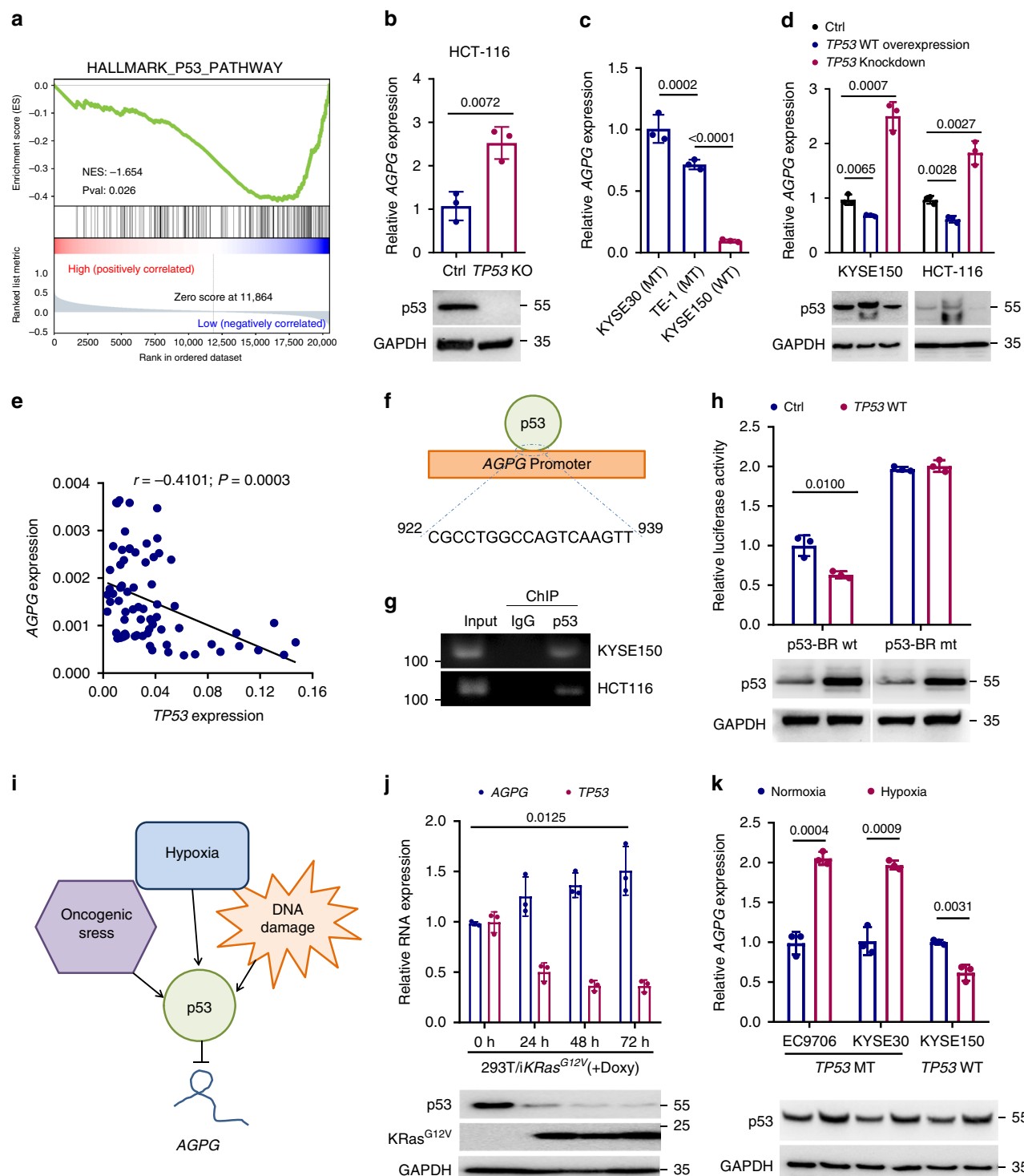

**Fig. 5 *AGPG* is transcriptionally regulated by p53. a** Pathway enrichment analysis suggested that *AGPG* was negatively correlated with p53. The pathway analysis was performed with GSEA method, which was based on an empirical permutation test procedure. **b** qPCR detection of *AGPG* expression in *TP53* KO and control HCT-116 cells. **c** qPCR detection of *AGPG* expression in ESCC cells with WT *TP53* (KYSE150) or MT *TP53* (KYSE30 and TE-1). **d** qPCR detection of *AGPG* expression in KYSE150 and HCT-116 cells with p53 upregulation or downregulation or control. **e** qPCR analysis showed that *AGPG* expression was negatively correlated with *TP53* expression in a cohort of ESCC patients with WT *TP53* (SYSUCC, Pearson's correlation analysis, $n = 72$). **f** The *AGPG* promoter contains a consensus p53-binding region. **g** ChIP assays showed that p53 bound to the *AGPG* promoter. **h** WT *TP53* overexpression diminished the transcriptional activity of *AGPG* in KYSE150 cells. **i** Schematic map of the regulatory network involving p53 and *AGPG*. **j** qPCR detection of *AGPG* expression in 293 T cells exposed to oncogenic stress (20 ng per ml doxycycline to induce ectopic KRas$^{G12V}$ expression) for 0, 24, 48, or 72 h. Doxy, doxycycline. **k** qPCR detection of *AGPG* expression in cells exposed to hypoxia or normoxia for 48 h. Data in **b–d**, **h**, **j**, **k** are representative of three independent experiments and presented as mean±S.D., $n = 3$ biologically independent samples, the *P* value was determined by a two-tailed unpaired Student's *t* test.

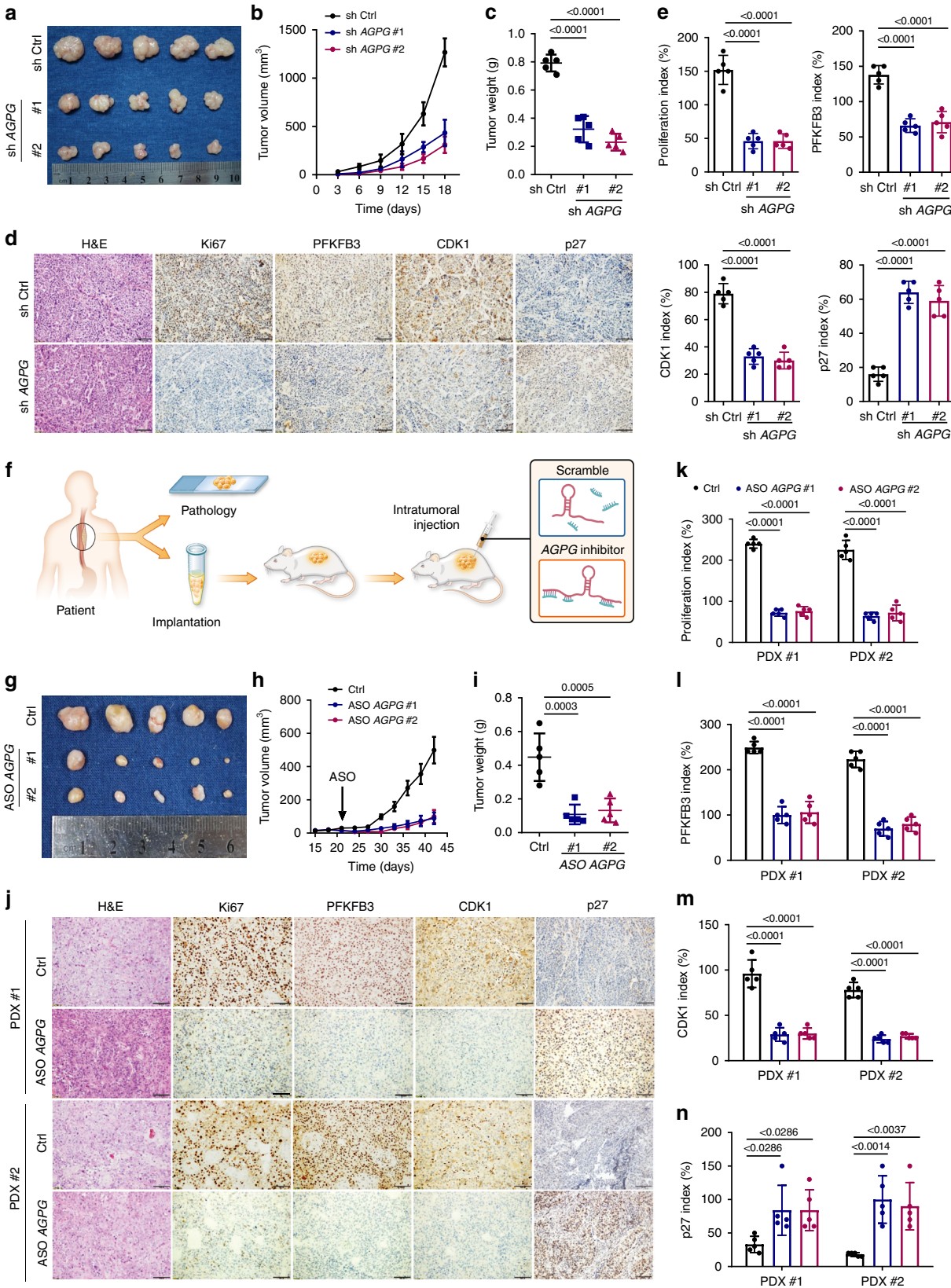

vectors were obtained from Kidan Biotechnology Co., Ltd. (Guangzhou, China). Briefly, *AGPG*-specific sgRNAs were designed to recognize two different sites of the *AGPG* gene, the location of the upstream target (>hg38_refGene_NR_002929_0 range=chr1:202861697-202861719) and the location of the downstream target (>hg38_refGene_NR_002929_6 range=chr1:202876356-202876378). To identify the effective sgRNAs, we transfected different sgRNA-Cas9 vectors into 293H cells, and then, the genomic DNA was extracted, Q5 PCR was performed and the PCR

products were subjected to sequencing. The results showed that *AGPG* sgRNA01 (CGGCGGGGCTGTTTCGTAAG) was effective among the upstream sgRNAs, and *AGPG* sgRNA02 (ATCAAGTGTCCTATATGCGT) was effective among the downstream sgRNAs. The PCR primer GCAACACCACGAATCCCAAC/TTGTC CCGCTCTGGAAACTC and the sequencing primer TTGTCCCGCTCTGGAAA CTC were used to detect the efficiency of *AGPG* sgRNA01. The PCR primer ACACTAGGCCATGCACCAA/GCCCACAGGCCAAATTCATTC and the

**Fig. 6 Effects of *AGPG* on tumor growth in vivo. a** *AGPG* knockdown inhibited cell-based xenograft growth in nude mice. **b**, **c** Statistical analysis of KYSE150 tumor volume and weight in nude mice. **d** Representative IHC images of randomly selected KYSE150 cell-based tumors from each group. Scale bar, 100 μm. **e** Quantification of IHC staining in KYSE150 cell-based tumors. **f** Graphic illustration of the intratumoral injection of in vivo-optimized *AGPG* inhibitor or control in the PDX models. **g** *AGPG* knockdown inhibited PDX growth in nude mice. The tumor tissues were from two ESCC patients (SYSUCC). **h**, **i** Statistical analysis of tumor volume and weight in the PDX #1 model. **j** Representative IHC images of randomly selected human-derived tumors from each group. Scale bar, 100 μm. **k–n** Quantification of IHC staining in human-derived tumors. Data in **b**, **c**, **e**, **h**, **i**, **k–n** are representative of three independent experiments and presented as mean±S.D., $n = 5$ mice per group, the $P$ value was determined by one-way ANOVA with Dunnett's multiple comparisons test. No adjustments were made for multiple comparisons.

---

sequencing primer GCCTCAGCCCACAGAGCTTA were used to detect the efficiency of *AGPG* sgRNA02. Thus, double sgRNA vectors were constructed using the sgRNAs mentioned above. After the construction of the vectors, the vectors were sequenced and compared with the target genes, which showed that the vectors were constructed correctly. ESCC cells were transfected with the pLV-U6-*AGPG* RNA sgRNA01-7SK-sgRNA02-EFS-hCas9-2A-Puro or pLV-U6-NC sgRNA01-7SK-NC sgRNA02-EFS-hCas9-2A-Puro expression vectors. The transfected cells were selected using puromycin (1 μg per ml) for 7 days. Isolated single colonies were expanded and subjected to detection of genomic deletions by PCR.

**Cell proliferation and cell cycle analysis**. Cell proliferation was measured with MTS (Qiagen, Hilden, German) according to the manufacturer's instructions[25]. Cell cycle analysis was performed with a cell cycle detection kit (KeyGen, China) according to the manufacturer's instructions. The cells were then analyzed with a Gallio flow cytometer (Beckman Coulter, CA, USA) and MultiCycle for Windows (Phoenix Flow Systems, CA, USA). The gating strategy for cell cycle analysis was provided in Supplementary Fig. 7.

**Western blot and qPCR analysis**. Western blot analysis was performed according to the manufacturer's instructions[7]. Cells or tissues were lysed in RIPA buffer. The protein concentrations were normalized with a BCA assay kit (Thermo Fisher Scientific, Carlsbad, CA, USA). Anti-GAPDH (1:1000, Cell Signaling Technology, Beverly, USA, 5174), anti-vinculin (1:1000, Cell Signaling Technology, 13901), anti-PFKFB3 (1:1000, Abcam, Cambridge, MA, USA, ab181861), anti-PFKFB3 (phospho S461)(1:1000, ab232498), anti-p27 (1:1000, Abcam, ab32034), anti-CDK1 (1:1000, Abcam, ab133327), anti-p53 (1:1000, Abcam, ab1101), anti-p21 (1:1000, Abcam, ab109199), anti-CDK3 (1:1000, Abcam, ab96847), anti-CDK6 (1:1000, Abcam, ab124821), anti-FLAG tag (1:1000, Cell Signaling Technology, 8146), anti-His tag (1:1000, Abcam, ab9108), anti-Cdc27 (1:1000, Abcam, ab10538), and anti-ubiquitin antibodies (1:1000, Cell Signaling Technology, 3933) were used in this study. Unprocessed images of the immunoblots are provided in the Supplementary Fig. 8. RNA levels were measured by qPCR analysis according to the manufacturer's instructions[51]. The specific primer sequences are listed in Supplementary Table 7.

**Determination of glycolytic activity**. Media samples were collected from cells cultured for 12 h or 24 h. Lactate concentrations were determined by a biosensor. ECAR was detected according to the XF Glycolysis Stress Test protocol on a Seahorse XFe24 Extracellular Flux Analyzer (Agilent Technologies, Santa Clara, CA, USA). [13]C-Labeled intracellular metabolites were detected as previously described[17]. Cells ($2 \times 10^7$) were incubated with 2 g per L $^{13}C_6$-labeled glucose (Cambridge Isotope Laboratories, CLM-1396-10) for 2 h. Metabolites were extracted, and those including at least one $^{13}$C atom were analyzed using an LC system equipped with a TripleTOF 5600 mass spectrometer (SCIEX, Framingham, MA, USA). The concentrations of lactate, ECAR, and $^{13}C_6$-labeled metabolites were normalized to cell number.

**RNA pull-down and RIP assays**. RNA was transcribed in vitro using a MEGAscript T7 Transcription Kit (Invitrogen) and biotinylated using a Pierce RNA 3′ End Desthiobiotinylation Kit (Thermo Scientific) according to the manufacturers' instructions. Cells were prepared using Pierce IP lysis buffer (Thermo Scientific). RNA pull-down assays were performed with a Pierce Magnetic RNA–Protein Pull-Down Kit. According to the manufacturer's instructions, biotinylated RNA was captured with streptavidin magnetic beads and then incubated with cell lysates or purified protein (20 μg) at 4 °C for 6 h before washing and elution of the RBP complex. The eluted proteins were subjected to MS analysis or western blotting. RIP assays were performed with a Magna RNA-binding protein immunoprecipitation kit (Millipore, Bedford, MA) according to the manufacturer's instructions. Negative control IgG, human anti-PFKFB3 antibody (1:20, Abcam, ab181861) and anti-FLAG tag antibody (1:20, Cell Signaling Technology, 8146) were used in this study. After proteinase K digestion, the immunoprecipitated RNAs were extracted, purified, and subjected to qPCR. RNA levels were normalized to the input (10%).

**MS2-tagged RNA affinity purification**. ESCC cells were cotransfected with pcDNA3.1-MS2/pcDNA3.1-MS2-*AGPG* and MCP-3xFLAG plasmids (OBiO

Technology, Shanghai, China). After 48 h, living cells were irradiated with 254 nm UV light at 400 mJ per cm². Then, the cells were lysed for 10 min on ice and centrifuged at $13,000 \times g$ for 10 min. FLAG-tagged proteins were immunoprecipitated with anti-FLAG M2 affinity gel (Sigma-Aldrich, USA). After three washes with low-salt wash buffer, agarose gels were boiled in loading buffer, and proteins were detected by western blotting analysis.

**CLIP-qPCR**. For sixteen hours before UVA exposure, 4-thiouridine (4-SU) was added to the ESCC cell culture medium at a final concentration of 100 μM. Then, the living cells were irradiated with 150 mJ per cm² UVA (365 nm) and lysed with NP-40 lysis buffer supplemented with protease inhibitors and 1 mM DTT. RNase T1 was added to the supernatant at a final concentration of 1 U per μl, and the samples were incubated at 22 °C for 15 min. Then, 40 μl of bead slurry was incubated with 10 μg of normal IgG or PFKFB3 antibody (Abcam, ab181861) for 2 h at 4 °C in NT2 buffer. The beads were washed, 1 mL of cell lysate was added to the antibody-coated Sepharose beads, and the mixtures were incubated for 3 h at 4 °C. The beads were washed, and the pellets were incubated with 20 units of RNase-free DNase I in 100 μl of NP-40 lysis buffer for 15 min at 37 °C. The pellets were washed and incubated with 0.1% SDS and 0.5 mg per ml proteinase K for 15 min at 55 °C. Then, the supernatants were collected, RNA was extracted, and qPCR analysis was performed.

**HITS-CLIP**. Cells were washed with ice-cold PBS three times, and UV crosslinking was performed with UV irradiation type C (254 nm) at 400 mJ per cm². The crosslinked cells were scraped off the plate and collected by centrifugation at $1000 \times g$ for 5 min. Cell lysis was performed in cold lysis buffer (1× PBS, 0.1% SDS, 0.5% NP-40 and 0.5% sodium deoxycholate) supplemented with a 1% RNase inhibitor (TaKaRa) and 2% protease inhibitor cocktail (Roche) for 30 min. Cell lysates were cleared by centrifugation at $8000 \times g$ for 10 min at 4 °C, and the supernatants were used for IP. For DNA digestion, RQ1 (Promega) was added to the lysate and incubated at 37 °C for 3 min. For IP, 600 μL of lysate was incubated with 15 μg PFKFB3 antibody (Abcam, ab181861) or control IgG antibody overnight at 4 °C. The immunoprecipitates were further incubated with protein A/G Dynabeads for 2–3 h at 4 °C. After the magnet was applied and the supernatant was removed, the beads were sequentially washed twice with wash buffer (1× PBS, 1% SDS, 0.5% NP-40 and 5% sodium deoxycholate), high-salt wash buffer (5× PBS, 1% SDS, 0.5% NP-40, and 5% sodium deoxycholate), and PNK buffer (50 mM Tris pH = 7.4, 10 mM MgCl₂, and 0.5% NP-40). The on-bead digestion was performed by adding MNase (Thermo), followed by incubation at 37 °C for 10 min. After the samples were washed with PNK buffer as described above, dephosphorylation and phosphorylation were performed with calf intestinal alkaline phosphatase (CIP, NEB) and polynucleotide kinase (PNK, NEB), respectively. The immunoprecipitated protein–RNA complex was eluted from the beads by heat denaturing and was resolved on a Novex 4–12% Bis-Tris precast polyacrylamide gel (Invitrogen). The protein–RNA complexes were cut from the gel[52], and RNA was extracted with TRIzol after digesting the proteins. The cDNA libraries were prepared using the KAPA Stranded RNA-Seq Library Preparation Kit (Kapa Biosystems). The cDNAs were purified and amplified, and PCR products corresponding to 200–500 bp were purified, quantified and stored at −80 °C before sequencing. For high-throughput sequencing, the libraries were prepared following the manufacturer's instructions and applied to the Illumina HiSeq X Ten system for 150 nt paired-end sequencing by ABlife, Inc. (Wuhan, China). The End1 3′ adapter 5′-AGATCGGAAGAGC-3′ and the End2 3′ adaptor 5′-AGATCGGAAGAGC-3′ were used in this study. We used HomeR to perform the motif analysis on the binding peaks obtained by the Piranha and CIMS analyses[19,20].

**Subcellular fractionation**. A Cytoplasmic & Nuclear RNA Purification Kit (Norgen Biotek Corp, Canada) was used to detect *AGPG* expression in cytoplasmic and nuclear fractions. According to the manufacturer's instructions, RNA was extracted from the cytoplasmic and nuclear fractions and subjected to qPCR. β-Actin was used as a cytoplasmic marker, and U6 was used as a nuclear marker.

**Fluorescence in situ hybridization (FISH) assay and immunofluorescence staining**. FISH assays were carried out with a lncRNA FISH Kit (RiboBio, Guangzhou, China). In brief, cells were fixed and permeabilized in PBS containing

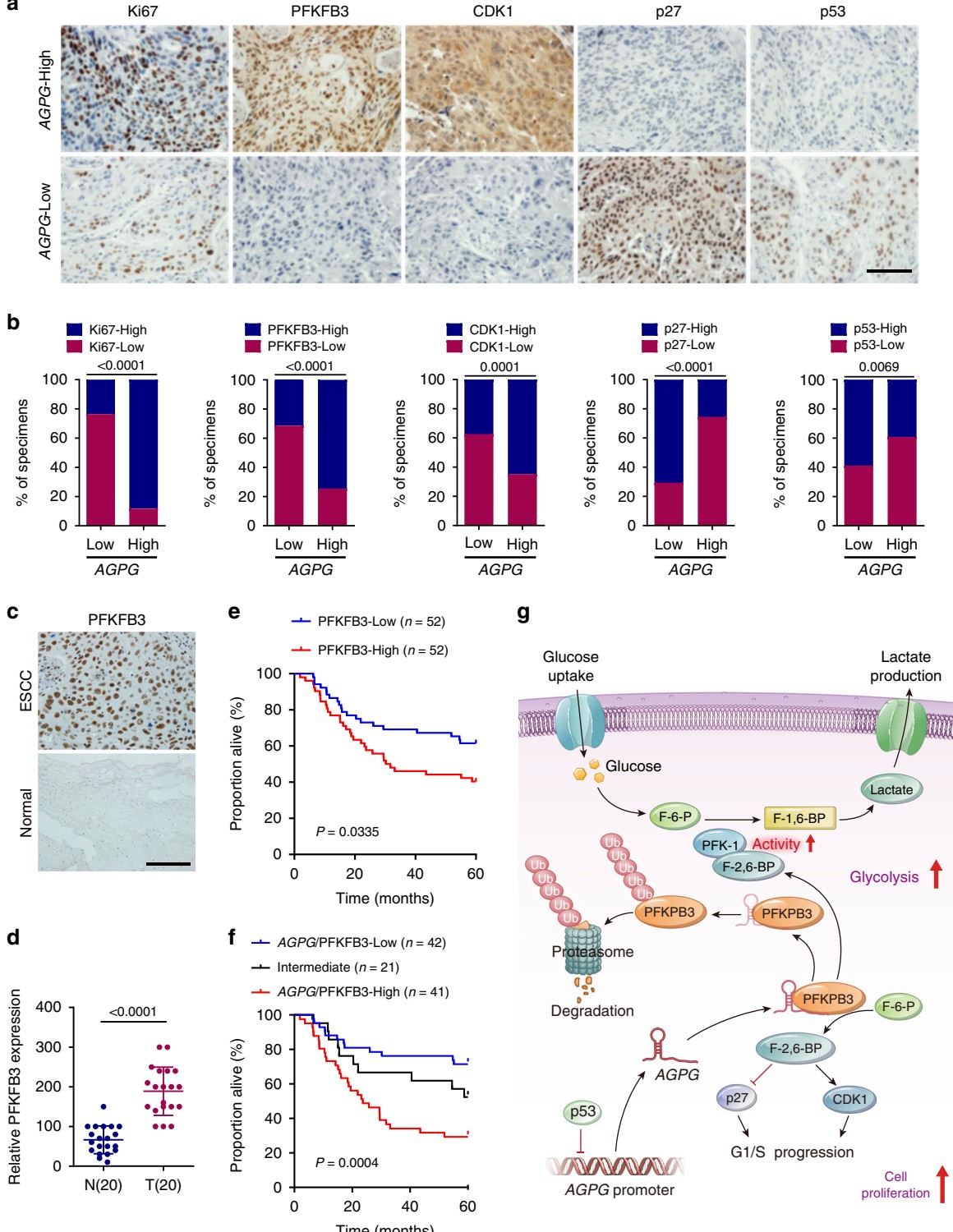

**Fig. 7 Clinical relevance of the p53-*AGPG*-PFKFB3 axis in ESCC. a** Representative IHC staining for Ki67, PFKFB3, CDK1, p27, and p53 in ESCC patients (SYSUCC, n = 104) with low or high *AGPG* expression. Scale bar, 100 μm. **b** Percentage of specimens with low or high Ki67, PFKFB3, CDK1, p27, and p53 expression in the low or high *AGPG* expression groups (SYSUCC, n = 104, Chi-square test, two-sided). **c, d** Representative IHC images and statistical analysis of PFKFB3 expression in ESCC and matched normal tissues (SYSUCC). Scale bar, 100 μm. Data are representative of three independent experiments and presented as mean±S.D., n = 20 cases, the P value was determined by a two-tailed unpaired Student's t test. **e** Kaplan–Meier analysis of the overall survival of ESCC patients (SYSUCC) with low (n = 52) or high (n = 52) PFKFB3 expression (log-rank test, two-sided). **f** Kaplan–Meier analysis of overall survival of ESCC patients (SYSUCC) with low (low expression of both *AGPG* and PFKFB3, n = 42), high (high expression of both *AGPG* and PFKFB3, n = 41) or intermediate (n = 21) *AGPG*/PFKFB3 expression (log-rank test, two-sided). **g** Graphical abstract showing that the lncRNA *AGPG* regulates glucose metabolism remodeling by affecting PFKFB3 stability.

0.5% Triton X-100. FISH probes were designed by RiboBio (Guangzhou, China). Hybridization was carried out overnight in a humidified chamber at 37 °C in the dark. All images were obtained with an Olympus FV1000 confocal microscope (Tokyo, Japan). 4′,6-diamidino-2-phenylindole and Cy3 channels were used to detect the signals. 18 S and U6 were used as the cytoplasmic and nuclear markers, respectively. Immunofluorescence staining was performed according to the manufacturer's instructions, and an anti-PFKFB3 antibody (1:150, Abcam, ab181861) was used in this study.

**ChIP, IP, and luciferase reporter assays**. ChIP assays were performed with a ChIP kit (Millipore). qPCR analysis was performed to detect the DNA fragments immunoprecipitated with p53. An anti-p53 antibody (1:20, Abcam, ab1101) was used in the ChIP assays. For the IP assays, an anti-PFKFB3 antibody (1:50, Abcam, ab181861) was used, and the immunoprecipitants were detected by western blotting. Luciferase reporter assays were performed according to the manufacturer's instructions (Promega, WI, USA). Firefly and Renilla luciferase activities were examined by the Dual-Luciferase Reporter Assay System, and firefly activity was normalized to Renilla activity.

**RNAScope ISH assay and IHC**. RNAScope ISH assays were carried out with an RNAScope 2.0 High Definition Assay Kit (Advanced Cell Diagnostics, Newark, CA, USA). A reactive score was obtained according to the percentage of positive cells and the staining intensity. For the IHC assays, staining and analysis were performed according to the manufacturer's instructions[53]. Anti-Ki67 (1:250, Abcam, ab15580), anti-PFKFB3 (1:250, Abcam, ab181861), anti-p27 (1:200, Abcam, ab32034), anti-CDK1 (1:250, Abcam, ab133327), and anti-p53 antibodies (1:200, Abcam, ab1101) were used. For quantification analysis, we evaluated the extent and intensity of all markers.

**Cell-based xenograft and PDX models**. ESCC cells ($2 \times 10^6$) expressing shCtrl or sh*AGPG* were injected subcutaneously into the dorsal flanks of 4-week-old female BALB/c nu/nu mice (five mice per group). Tumor growth was monitored every 3 days after transplantation using calipers. To generate PDX models, fresh ESCC tumor samples from patients were immediately inoculated subcutaneously into both flanks of nude mice. When the successfully established PDXs (P1) reached ~ 500 mm[3], the tumors were transplanted to other mice (P2). Eventually, the mice bearing P3 grafts were used to examine the therapeutic effects of *AGPG* inhibitor. Twenty-one days after transplantation, we began to perform intratumoral injections of scrambled or in vivo-optimized *AGPG* inhibitor (5 nmol per injection, RiboBio, Guangzhou, China) every 3 days. When the study finished, the mice were anesthetized, and the tumor volume and weight were measured. All tissues from the cell-based xenografts or PDXs underwent further pathological analysis.

**Statistics and reproducibility**. All experiments were carried out at least three times, for RNAScope ISH, FISH, immunofluorescence staining, IHC, and western blot assays, representative images are shown. The results are presented as the mean ±S.D. of at least three independent experiments after analysis by Student's $t$ test or one-way ANOVA using GraphPad Prism 8.0.1 (GraphPad, La Jolla, CA, USA). Relative gene expression was analyzed using the $2^{-\Delta Ct}$ or $2^{-\Delta\Delta Ct}$ method. Correlations between *AGPG* levels and PFKFB3 expression were analyzed with Pearson's correlation analysis. Survival analyses were performed using the Kaplan–Meier method and assessed using the log-rank test. All the statistical tests were two-sided, $P < 0.05$ was considered statistically significant.

**Study approval**. The study was approved by the Medical Ethics Committee of SYSUCC. Written informed consent was obtained from the patients who provided samples. The animal studies were approved by the Institutional Animal Care and Use Committee of Sun Yat-Sen University.

**Reporting summary**. Further information on research design is available in the Nature Research Reporting Summary linked to this article.

## Data availability

Expression and survival analyses for lncRNAs in ESCC were performed using data obtained from TCGA[54] (http://www.cbioportal.org/publicportal/). DESeq2 (version 3.10) was employed for differential expression analysis[55]. The pathway analysis was performed with GSEA (version 4.0.3)[56], whereas JASPAR software (version 2018) predicted the p53-binding sequence[57]. The posttranslational modification data of PFKFB3 were obtained from the dbPAF (version 1.0)[58] and CPLM databases (version 1.0)[59] and visualized by IBS software (version 1.0.3)[60]. The CLIP-Seq data set is available at NCBI Sequence Read Archive (SRA) under BioProject PRJNA591321 (https://www.ncbi.nlm.nih.gov/bioproject/591321). For CLIP-seq analysis, induced HomeR (homer2 version) was used to perform the motif analysis on the binding peaks obtained by the Piranha (version 1.2.1) and CIMS (version 1.1.3) analyses. All the other data supporting the findings of this study are available within the article and its Supplementary Information Files or from the corresponding authors upon reasonable request.

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

## Acknowledgements

This research was supported by the Natural Science Foundation of Guangdong Province (2019A1515010233), Fundamental Research Funds for the Central Universities (19ykyjs67), Natural Science Foundation of China (81972239, U1601229), Program for Guangdong Introducing Innovative and Entrepreneurial Teams (2017ZT07S096), Guangdong Esophageal Cancer Institute Science and Technology Program (M201707), Young Teacher Training Program of Sun Yat-Sen University (17ykpy78), and CAMS Innovation Fund for Medical Sciences (2019-I2M-5-036).

## Author contributions

R.-H.X., H.-Q.J., and J.L. designed the study. J.L., Q.-N.W., Y.-X.L., C.-W.W., L.M., Y.W., Z.-X.W., M.-M.H., Y.J., and C.R. collected the data. R.-H.X., H.-Q.J., J.L., Z.-X.L., Q.-N.W., Y.-X.L., C.-W.W., L.M., and Z.-X.W. analyzed and interpreted the data. R.-H.X., H.-Q.J., J.L., Z.-X.L., Q.-N.W., Y.-X.L., and C.-W.W. wrote the manuscript. R.-H.X., J.L., Z.-X.L., Q.-N.W., Y.-X.L., D.-S.W., D.-L.C., H.-Y.P., D.X., M.-S.Z., L.F., B.L., A.L., P.H., D.-X.L., and H.-Q.J. revised the manuscript. R.-H.X., H.-Q.J., J.L., Z.-X.L., Q.-N.W., Y.-X.L., C.-W.W., and L.M. performed the statistical analysis. All authors reviewed the manuscript and approved the final version.

## Competing interests

The authors declare no competing interests.
