## [Peer Review File · Nature Communications]

Reviewers' comments:

Reviewer #1 (Remarks to the Author):

In this manuscript, Jia Liu and co-author identified a new lncRNA AGPG promote glycolysis and cell proliferation in esophageal squamous cell carcinoma. Mechanistically, the authors revealed that AGPG increase the PFKFB3 protein stability through binding with the C-terminal of PFKFB3 and subsequently activates glycolytic flux and promotes cell cycle progression. The authors also identified P53 can repress the AGPG expression. Although the authors may have been eager to convey this idea, the lack of novel mechanistic insights, precludes this manuscript's publication on Nature Communications. What do they tell us about the functions of lncRNA that we could not have anticipated? Even after weeding through the multiple panels and approaches, I do not see a broad message that would entice the Nature Communications reader.

Major concerns

1. The experiment in which the authors attempted to mapping AGPG and PFKFB3 interaction seem rather naïve without any effort to carry out more convincing experiments. First, I request PFKFB3 CLIP experiment to confirm the regions of AGPG responsible for PFKFB3 binding. Second, this key mechanistic step should be shown by functional rescue experiments to confirm that it indeed happens and in the direct manner postulated. Rescue experiments overexpressing wild-type AGPG or the PFKFB3-binding deficient mutant in AGPG CRISPR knockout cells are essential.

3. The authors' model is that AGPG functions essentially to modulate PFKFB3 ubiquitination. A basic test of this model would be to assess the absolute copy number of the AGPG versus that of PFKFB3. lncRNA copy numbers are often low, so are there sufficient copies to make the suggested stoichiometric complexes? AGPG lncRNA should be modified to include an MS2 binding site and expressed in cells to show that the same proteins are associated with AGPG lncRNA in vivo?

4. It is important for the authors to define what's the underlying molecular mechanism for how AGPG can prevent PFKFB3 ubiquitination? What are the E3 and E2 enzymes responsible for PFKFB3 ubiquitination? Will AGPG and E3/E2 compete for binding with PFKFB3? Technically, the full size of FLAG IB panels in Figure 4i and j should be shown to reveal the ubiquitinated PFKFB3. Also evidence of endogenous PFKFB3 K302 ubiquitination must be provided.

5. PFKFB3 contains 2 functional domains: kinase and bisphosphatase, located at N-terminal and C-terminal, respectively. PFKFB3 does not have any RNA binding motifs or domains. What's the potential explanation of AGPG binding to C-terminal bisphosphatase domain of PFKFB3? Is there other binding partners? The mass spec data should also be provided. Conceptually, I fail to see whether AGPG would affect PFKFB3 enzymatic activity by binding to its C-terminal domain. This really is a crucial point that must be explored with some detail and accuracy.

Minor concerns

1. Lactate production will increase as the cells growing. The screening criteria (cell viability and lactate production) used here are biased because cell viability will definitely affect lactate production rate. Reduced lactate production could just be a side effect of less viable cells.

2. Patient samples: how low expression and high expression are determined?

3. PFKFB3 has been shown regulating of glycolysis, cell viability and proliferation. Like PFKFB3, AGPG also regulates cell proliferation, cell cycle and glycolysis. Is this dependent on PFKFB3? Have you tried testing AGPG effect on PFKFB3 knockout cell lines?

4. Does AGPG affect PFKFB3 gene expression? Does AGPG affect PFKFB3 subcellular location?

Nuclear expressed PFKFB3 had no effect on glucose metabolism. The regulation of glycolysis by PFKFB3 is mainly in the cytoplasm and the control of cell cycle is in the nucleus. Where is AGPG associated with PFKFB3, cytosol or nucleus?

5. Fig. 2E: why all the samples have the same G2/G1 ratio?

6. Since AGPG is transcriptionally regulated by p53, it's surprising to see wild-type p53 and mutant p53 has completely opposite effects on AGPG expression (Fig. 5k). While WT p53 acts predominantly as a transcription factor, a number of the described gain-of-function activities of mutant p53 are mediated through nontranscriptional process. It has been shown that WT p53 is dependent on 4 hydrophobic amino acids in its N terminal for transactivation activity. More evidence needs to be provided to confirm AGPG as a p53 target.

7. In Fig1, The authors according the TCGA database find out 50 top lncRNAs to do the siRNAs screening. For the lactate measure, did the concentration of lactate was normalized to the cell number? For the siRNAs screening, how to exclude siRNA off target affect? Authors should explain it in detail.

8. For the Fig1B, the authors need use more cell lines to confirm the siRNAs screening results so that to demonstrate AGPG specific regulation ESCC cells glycolysis.

9. Some of results showed not consistent in this study. Such as in Fig2I, EC9706 and KYSE30 cells glycolysis activity about 100 mpH/min in control group, but in Fig4M why glycolysis activity change to about 50 mpH/min in the same cells and same time overexpressed PFKFB3. Authors need explain what the reasons can lead to this difference.

10. In Fig3, the authors need supply RNA pulldown mass spectrometry raw data. In addition to PFKFB3 binding to AGPG, are there other proteins that are also binding to AGPG?

11. The authors claimed AGPG mainly located in nuclear. For the animal and cellular models, why not to use the Antisense Oligonucleotides (ASO) to knockdown AGPG? Because compare to shRNAs, ASO is better to knockdown nuclear located RNAs.

12. In Fig2K, the sh Ctrl group need normalized to 1.

Reviewer #2 (Remarks to the Author):

This study identifies a lncRNA AGPG that is important for glycolysis and tumor development in esophageal squamous cell carcinoma (ESCC). Mechanistically, they show that AGPG can interact with the glycolytic enzyme PFKFB3 and somehow this interaction is important to maintain PFKFB3 protein stability.

In recent years many lncRNAs have been identified to regulate glycolysis or cancer metabolism through diverse array of mechanisms. This study identified one more such lincRNA. The novelty and broad impact of the study is considered to be relatively moderate. In addition, there are several significant weaknesses that prevent its publication in Nature Communications in its current format. The specific comments are:

1. Based on the data from Fig. 1, lncRNA AGPG is mainly localized in nucleus. There are wide concerns on using shRNA to study nucleus-localized lncRNAs, because shRNA mainly works for cytoplasm-localized RNA, such as mRNAs encoding proteins. The authors showed that they can rescue the phenotypes from shRNA studies by restoring the cDNA of this lncRNA. This approach is commonly used in studying protein-coding genes by restoring a cDNA with silent mutation on the

corresponding sites in shRNA. However, it would be difficult to use this approach for lncRNA (or any non-coding gene). So it remains unclear how they can restore the expression of AGPG because overexpressed AGPG should still be targeted by its shRNA. The authors need to confirm their key findings with CRISPR technology, which has been commonly used in studying lncRNAs now.

2. Another major weakness of this study is the lack of detailed mechanism by which AGPG regulates PFKFB3 protein stability. It is hard to imagine that, without any other protein partner, this lncRNA is sufficient to stabilize a protein. Does AGPG promote PFKFB3 protein stability through regulating another E3 ubiquitin ligase or DUB? The authors need to present more mechanistic data to address this question.

3. The study showed that AGPG deficiency resulted in defects in glycolysis, cell proliferation, and tumor growth, with decreased PFKFB3 levels. However, whether AGPG regulates these cellular functions THROUGH PFKFB3 remains unclear. It is still possible that AGPG regulates cancer metabolism through other targets. The authors need to examine whether restoring PFKFB3 K302A (a mutant that is resistant to AGPG regulation) in AGPG deficient cells would at least partially rescue defects in glycolysis, cell proliferation, and tumor development caused by AGPG deficiency.

4. The study used delta T5 of AGPG as a non-binding mutant to PFKFB3. However, there is no data to show that delta T5 is indeed a non-binding mutant. Fig. 3G and 3H showed that T1, T2, T3, or T4 does not interact with PFKFB3, but they never examine the binding between T1+2+3+4 (which is delta T5) and PFKFB3.

5. For patient survival analyses, it is unclear why some analyses choose 50%-50% to separate low vs high expression (which is the standard way for such analyses), whereas other data (such as Fig. 1E) choose other ways for separation. All analyses should be conducted using 50%-50% separation.

6. In Fig. 6F, they delivered AGPG siRNA in vivo to PDX models, and showed this impacted on tumor development. However, there is no data to document that siRNA indeed decreased AGPG expression in tumors. They need to show this.

Minor points:

1. Fig. 1G shows that, while AGPG is overexpressed in some cancers, in many other cancers there is no difference or its level is even significantly decreased in several cancers, such as KIRP and KIRP. The authors need to comment on this.

2. The study identified PFKFB3 as a AGPG-interacting protein based on MS analysis. They need to include the list of the proteins (including peptide numbers) identified from MS.

3. Page 14 states that "However, very few lncRNAs have been implicated in cancer metabolism regulation, and the underlying mechanisms remain poorly understood." This clearly is an overstatement. In recent years, many lncRNAs have been identified to regulate cancer metabolism. There are now many reviews to discuss and summarize this research topic. At the best, this study identified the first lncRNA to directly regulate PFKFB3.

Response to the reviewers: Thank you very much for reviewing our manuscript. We appreciate your critiques and suggestions, which were very constructive and helpful for strengthening our manuscript. We have addressed all your concerns in the revised manuscript and have included point-by-point responses in this letter. All changes are identified by page and line location and noted by highlighting or strikethrough in the revised manuscript with tracked changes. Our detailed responses are as follows.

Reviewer #1 (Remarks to the Author):

In this manuscript, Jia Liu and co-author identified a new lncRNA *AGPG* promote glycolysis and cell proliferation in esophageal squamous cell carcinoma. Mechanistically, the authors revealed that *AGPG* increase the PFKFB3 protein stability through binding with the C-terminal of PFKFB3 and subsequently activates glycolytic flux and promotes cell cycle progression. The authors also identified P53 can repress the *AGPG* expression. Although the authors may have been eager to convey this idea, the lack of novel mechanistic insights, precludes this manuscript's publication on Nature Communications. What do they tell us about the functions of lncRNA that we could not have anticipated? Even after weeding through the multiple panels and approaches, I do not see a broad message that would entice the Nature Communications reader.

Major concerns

1. The experiment in which the authors attempted to mapping *AGPG* and PFKFB3 interaction seem rather naïve without any effort to carry out more convincing experiments. First, I request PFKFB3 CLIP experiment to confirm the regions of *AGPG* responsible for PFKFB3 binding. Second, this key mechanistic step should be shown by functional rescue experiments to confirm

that it indeed happens and in the direct manner postulated. Rescue experiments overexpressing wild-type *AGPG* or the PFKFB3-binding deficient mutant in *AGPG* CRISPR knockout cells are essential.

Response: Thank you for the suggestion. First, following your suggestion, we performed crosslinking immunoprecipitation and qPCR (CLIP-qPCR), an improved method for the isolation of lncRNA segments bound by various proteins, to further confirm the regions of *AGPG* responsible for PFKFB3 binding¹. Consistent with our previous results, the T5 fragment of *AGPG* was identified as the main region responsible for binding PFKFB3 (Figure 3H, Page 9, Line 165). Second, we constructed *AGPG* knockout (KO) cell lines using a CRISPR/Cas9-based strategy (Figure S3F). To confirm that the T5 fragment is required for *AGPG* function in interacting with and regulating PFKFB3, we performed rescue experiments in which we overexpressed wild-type (WT) *AGPG* or a mutant lacking the T5 fragment (*AGPG* Δ T5) in *AGPG* KO cells. As expected, overexpression of full-length *AGPG* (*AGPG* FL), but not *AGPG* Δ T5, was sufficient to prevent the phenotypes observed after *AGPG* KO, including glycolytic reprogramming, cell proliferation and cell cycle regulation (Figures 3J-3K, S3G-S3I, Page 9, Line 168).

2. The authors' model is that *AGPG* functions essentially to modulate PFKFB3 ubiquitination. A basic test of this model would be to assess the absolute copy number of the *AGPG* versus that of PFKFB3. lncRNA copy numbers are often low, so are there sufficient copies to make the suggested stoichiometric complexes? *AGPG* lncRNA should be modified to include an MS2 binding site and expressed in cells to show that the same proteins are associated with *AGPG* lncRNA *in vivo*?

Response: Thank you for the suggestion. The absolute copy number analysis performed as suggested by the reviewer showed that there were ~400–700

AGPG molecules per cell versus ~4400–7400 PFKFB3 molecules per cell in the EC9706 and TE-1 ESCC cell lines (Figure S3D, Page 8, Line 153). Even if lncRNAs have relatively lower copy numbers, they can play important regulatory roles in cells^{2, 3}. The expression abundance of *AGPG* is not low among lncRNAs, and our results indicated that there are sufficient *AGPG* copies in ESCC cells and that the endogenous *AGPG*-PFKFB3 interaction might occur frequently^{3, 4}. As suggested by the reviewer, MS2-tagged RNA affinity purification (MTRAP) and western blotting were performed to further characterize the interaction between *AGPG* and PFKFB3 *in vivo*⁵; compared with the negative control, coexpression of the MS2-*AGPG* and MCP-3xFLAG plasmids led to a significant enrichment of PFKFB3, demonstrating that PFKFB3 specifically binds to *AGPG* (Figure 3D, Page 7, Line 143).

3. It is important for the authors to define what's the underlying molecular mechanism for how *AGPG* can prevent PFKFB3 ubiquitination? What are the E3 and E2 enzymes responsible for PFKFB3 ubiquitination? Will *AGPG* and E3/E2 compete for binding with PFKFB3? Technically, the full size of FLAG IB panels in Figure 4i and j should be shown to reveal the ubiquitinated PFKFB3. Also evidence of endogenous PFKFB3 K302 ubiquitination must be provided.

Response: Thank you for the suggestion. As reported previously⁶, PFKFB3 is degraded through the ubiquitin-proteasome pathway. PFKFB3 contains a KEN box, which targets proteins for ubiquitylation by APC/C. It has been reported that PFKFB3 is subject to degradation involving APC/C-Cdh1, a cell cycle-regulated E3 ubiquitin ligase. Our data suggest that *AGPG* is required for PFKFB3 stabilization because it prevents PFKFB3 ubiquitination. To further elucidate the mechanism by which *AGPG* blocks PFKFB3 ubiquitination, we performed coimmunoprecipitation (coIP) assays to identify whether *AGPG* affects the interaction between PFKFB3 and APC/C. Active APC/C was immunoprecipitated from ESCC cells using an anti-Cdc27 antibody (Abcam,

ab10538). As shown in Figure 4H (Page 10, Line 195), *AGPG* KO significantly increased the interaction between PFKFB3 and Cdc27, suggesting that *AGPG* blocks the binding of Cdc27 to PFKFB3. Collectively, *AGPG* specifically binds to PFKFB3 and blocks its interaction with APC/C, which inhibits APC/C-mediated PFKFB3 ubiquitination and subsequent degradation. The complete, uncropped FLAG IB images of the immunoblots in Figure 4G and 4 I are shown in the supplementary data (Page 13). Evidence for endogenous PFKFB3 K302 ubiquitination was obtained from the dbPAF⁶ and CPLM databases⁸, and the results were visualized by IBS software⁹. Among these ubiquitinome profiling studies, 4 putative sites of lysine ubiquitination, including K302, were detected by the mass spectrometric analysis¹⁰⁻¹⁴.

4. PFKFB3 contains 2 functional domains: kinase and bisphosphatase, located at N-terminal and C-terminal, respectively. PFKFB3 does not have any RNA binding motifs or domains. What's the potential explanation of *AGPG* binding to C-terminal bisphosphatase domain of PFKFB3? Is there other binding partners? The mass spec data should also be provided. Conceptually, I fail to see whether *AGPG* would affect PFKFB3 enzymatic activity by binding to its C-terminal domain. This really is a crucial point that must be explored with some detail and accuracy.

Response: We apologize for not providing a clear explanation and evidence. In our study, we performed a variety of *in vitro* and *in vivo* experiments to study the interaction between the lncRNA and the protein, including RNA pull-down, RIP, CLIP and MTRAP assays. Consistently, we demonstrated that *AGPG* directly associated with PFKFB3, which implies that PFKFB3 is a newly discovered noncanonical RBP (Page 15, Line 289). Canonical RNA-binding proteins (RBPs) generally contain RNA-binding regions, but recent studies have indicated that hundreds of new RBPs lack known RNA-binding domains, indicating the complexity and diversity of RNA-protein complexes¹⁵. A large

number of RBPs are gradually being discovered using various experimental methods. Therefore, key proteins could be missed by predicting whether a protein is an RBP based solely on protein sequence and structural information. Mass spectrometry data are provided in supplementary Table S7 (Page 7, Line 140), and PFKFB3 was identified as an *AGPG*-associated protein.

PFKFB3 contains 2 functional domains, an N-terminal kinase domain and a C-terminal bisphosphatase domain. The bifunctional isoenzyme encoded by the PFKFB3 gene has the highest ratio of kinase:phosphatase activity, which sustains high glycolytic rates¹⁶. Regarding PFKFB3 enzymatic activity, we demonstrated that *AGPG* KO had no effect on PFKFB3 S461 phosphorylation (Figure S4G), which is widely reported to be a key regulator of PFKFB3 enzyme activity^{14, 17, 18}; thus, we speculate that *AGPG* has little effect on PFKFB3 enzyme activity (Page 10, Line 191).

Minor concerns

1. Lactate production will increase as the cells growing. The screening criteria (cell viability and lactate production) used here are biased because cell viability will definitely affect lactate production rate. Reduced lactate production could just be a side effect of less viable cells.

Response: We apologize for not providing a clear explanation. Lactate production will certainly increase as cells grow; to exclude this bias, all lactate concentrations were normalized to cell number^{14, 19}. In our revised manuscript (Page 19, Line 388), we specifically emphasized this issue as follows: “The concentrations of lactate, ECAR, and ¹³C6-labeled metabolites were normalized to cell number.” Therefore, our screening criteria enabled the identification of lncRNAs that affect glucose metabolism and cell proliferation.

2. Patient samples: how low expression and high expression are determined?

Response: We apologize for not providing a clear explanation. In this study, we categorized gene expression as low or high in comparison with the median value: if the expression level was higher than the median, it was classified as high, whereas if it was lower than the median, it was low. We have added this information on Page 6, Line 102.

3. PFKFB3 has been shown regulating of glycolysis, cell viability and proliferation. Like PFKFB3, *AGPG* also regulates cell proliferation, cell cycle and glycolysis. Is this dependent on PFKFB3? Have you tried testing *AGPG* effect on PFKFB3 knockout cell lines?

Response: Thank you for the suggestion. *AGPG* KO in ESCC cells strikingly inhibited glycolysis, proliferation and cell cycle progression. Then, we performed a series of rescue experiments to investigate the role of PFKFB3 in *AGPG*-mediated cell proliferation and glycolytic reprogramming. As shown, PFKFB3 overexpression could reverse the inhibition of glycolysis, cell proliferation and cell cycle progression by *AGPG* KO (Figures 4K-4M, S4K-S4L, Page 11, Line 213). Following the reviewer's suggestion, we also ascertained the effect of *AGPG* on PFKFB3 KO cell lines. After PFKFB3 KO, *AGPG* KO had mild effects on aerobic glycolysis, proliferation and cell cycle regulation in ESCC cells (Figure S4M-S4O, Page 11, Line 218). Collectively, these data suggest that the regulatory roles of *AGPG* in cell proliferation and glycolytic reprogramming are mainly dependent on PFKFB3.

4. Does *AGPG* affect PFKFB3 gene expression? Does *AGPG* affect PFKFB3 subcellular location? Nuclear expressed PFKFB3 had no effect on glucose metabolism. The regulation of glycolysis by PFKFB3 is mainly in the cytoplasm and the control of cell cycle is in the nucleus. Where is *AGPG* associated with

PFKFB3, cytosol or nucleus?

Response: Thank you for noting this issue. Our results showed that *AGPG* KO did not affect PFKFB3 mRNA levels or subcellular location (Figure S4E-S4F, Page 10, Line 190). Accumulating studies have suggested that PFKFB3 is a vital regulator of glycolysis, and some researchers have also reported its role in regulating cell proliferation via glycolysis-independent CDK activities in the nucleus^{14, 19}. However, to the best of our knowledge, due to the complexity of the intracellular metabolic regulatory network, there is insufficient evidence that nuclear PFKFB3 is not involved in glycolytic regulation. In addition, our immunofluorescence colocalization analysis showed that *AGPG* and PFKFB3 colocalized mainly in the nucleus but also in the cytoplasm, which suggests that the *AGPG*-PFKFB3 complex may play a role in both the nucleus and cytoplasm (Figure 3E, Page 8, Line 147). Therefore, consistent with our results, the *AGPG*-PFKFB3 complex is involved in glucose metabolism and cell cycle regulation.

5. Fig. 2E: why all the samples have the same G2/G1 ratio?

Response: Thank you for noting this issue. In the flow cytometric measurement of cellular DNA content, the DNA distribution usually has two peaks. The second peak, which corresponds to the 4C DNA content of cells in G2 and M, is often located at a value lower than twice that of the 2C DNA peak, which contains cells in G1. We analyzed all the samples and found that they had an average G2/G1 peak position ratio of 1.9, but each sample had a slightly different value. We apologize for not providing the specific G2/G1 ratio of each sample. Therefore, the specific G2/G1 ratio is provided in the revised Figure 2E.

6. Since *AGPG* is transcriptionally regulated by p53, it's surprising to see

wild-type p53 and mutant p53 has completely opposite effects on *AGPG* expression (Fig. 5k). While WT p53 acts predominantly as a transcription factor, a number of the described gain-of-function activities of mutant p53 are mediated through nontranscriptional process. It has been shown that WT p53 is dependent on 4 hydrophobic amino acids in its N terminal for transactivation activity. More evidence needs to be provided to confirm *AGPG* as a p53 target.

Response: Thank you for the suggestion. The p53 transcriptional response involves both the activation and repression of numerous genes. While p53 is known to transcriptionally activate numerous genes, the mechanisms by which p53 leads to gene repression have remained elusive. As reported previously, each functional domain of p53 has a different biological activity. For example, residues 22-23 in the N-terminus of p53 are involved in transcriptional activation²⁰, and residues 339-346 in the C-terminus of p53 might play a role in transcriptional repression²¹. p53 can directly repress transcription by binding p53 response elements in, for example, the CD44 or Per2 promoter²¹. This interaction may represent a key regulatory link in the p53-mediated cellular response. In our study, residues 922-939 in the *AGPG* promoter were found to contain a conserved p53-binding sequence (Figure 5F), which was identified as a p53-binding region (p53-BR) by chromatin immunoprecipitation (ChIP) assays (Figures 5G, S5A). Consistently, the transcriptional activity of luciferase reporters containing an intact p53-BR (p53-BR wt) was markedly weaker than that of reporters with the p53-BR deleted (p53-BR mt). Moreover, cotransfection of WT p53 selectively decreased the transcriptional activity of reporters with an intact p53-BR (Figures 5H, S5B). These methods are widely used to identify p53 targets^{23, 24}. Collectively, our data support *AGPG* as a p53 target with a functional p53-BR.

7. In Fig1, The authors according the TCGA database find out 50 top lncRNAs

to do the siRNAs screening. For the lactate measure, did the concentration of lactate was normalized to the cell number? For the siRNAs screening, how to exclude siRNA off target affect? Authors should explain it in detail.

Response: Thank you for the suggestion. As mentioned above, all lactate concentrations were normalized to cell number. In our revised manuscript (Page 19, Line 388), we emphasized this issue as follows: “The concentrations of lactate, ECAR, and ¹³C6-labeled metabolites were normalized to cell number.” We apologize for not providing a clear explanation of the siRNA screening. Therefore, in our revised manuscript (Page 5, Line 84), we explained this issue as follows: “For the siRNA screening, the siRNA library was designed with the SMARTselection algorithm to ensure high-efficiency silencing. These siRNAs also contained the proprietary ON-TARGETplus dual-strand chemical modification to ensure optimal strand loading and to disrupt microRNA-like seed activity, thereby reducing off-target effects²⁵.” To further exclude the siRNAs off-target effects and confirm the function of *AGPG*, we designed multiple shRNA sequences to knockdown *AGPG* and then verified its effects on cell proliferation and glucose metabolism. As you suggested, we also performed CRISPR technology to confirm our findings.

8. For the Fig1B, the authors need use more cell lines to confirm the siRNAs screening results so that to demonstrate *AGPG* specific regulation ESCC cells glycolysis.

Response: Thank you for the suggestion. Following the reviewer’s suggestion, we transfected the siRNA library into KYSE150 cells and examined cell viability and lactate production. Consistent with our previous screening results, *AGPG* knockdown significantly decreased cell viability and lactate production among the lncRNA candidates (Figure S1B).

9. Some of results showed not consistent in this study. Such as in Fig2I, EC9706 and KYSE30 cells glycolysis activity about 100 mpH/min in control group, but in Fig4M why glycolysis activity change to about 50 mpH/min in the same cells and same time overexpressed PFKFB3. Authors need explain what the reasons can lead to this difference.

Response: Thank you for noting this issue. In our study, glycolysis activity was detected with a Seahorse XFe24 Extracellular Flux Analyzer. The system measures glycolysis by measuring the rate of extracellular acidification (ECAR). Therefore, it is very sensitive to changes in cell number and cellular metabolism. In our previous Fig2I and Fig4M, different seeding numbers between the two experiments may cause some fluctuations in the detected ECAR values. Thus, we used similar seeding numbers in our revised manuscript (Figure 3J, S3G-S3H, 4K-4L, S4K) to minimize fluctuations in ECAR values between experiments. Therefore, the effects of *AGPG* and PFKFB3 on glycolysis could be more accurately identified.

10. In Fig3, the authors need supply RNA pulldown mass spectrometry raw data. In addition to PFKFB3 binding to *AGPG*, are there other proteins that are also binding to *AGPG*?

Response: Thank you for the suggestion. We apologize for not providing the RNA pull-down mass spectrometry raw data. These data are now provided in supplementary Table S7 (Page 7, Line 140), and PFKFB3 was identified as an *AGPG*-associated protein. We focused on PFKFB3 because it is a known regulator of metabolism and cell proliferation. Also, there were a number of proteins bound non-specifically to *AGPG*, as noted by their binding to beads or the antisense control. Therefore, we failed to show that *AGPG* specifically interacted with other proteins.

11. The authors claimed *AGPG* mainly located in nuclear. For the animal and cellular models, why not to use the Antisense Oligonucleotides (ASO) to knockdown *AGPG*? Because compare to shRNAs, ASO is better to knockdown nuclear located RNAs.

Response: Thank you for the suggestion. Our previous results showed that shRNA was effective (Figure S2A); therefore, we performed functional experiments with shRNA, which has been used in some nuclear-localized lncRNA studies²⁶. However, following the reviewer's suggestions, we also confirmed our findings with CRISPR/Cas9 technology. As shown in Figures 3J-3K and S4F-S4I, *AGPG* KO in ESCC cells strikingly inhibited glycolysis, proliferation and cell cycle progression, further confirming that *AGPG* is essential for glycolytic reprogramming, proliferation and cell cycle progression in ESCC. And the main component of the *in vivo*-optimized *AGPG* inhibitor used in the PDX model was actually ASO²⁷, we apologize for not providing an accurate description before. We have added this information on Page 13, Line 257. We also detected the *AGPG* expression in PDX tumor tissues by q-PCR analysis, as shown in Figure S6G, *in vivo*-optimized *AGPG* inhibitor significantly inhibited the *AGPG* expression in the tumors.

12. In Fig2K, the sh Ctrl group need normalized to 1.

Response: Thank you for noting this issue. In the revised Figure 2K, the shCtrl group was normalized to 1.

Reviewer #2 (Remarks to the Author):

This study identifies a lncRNA *AGPG* that is important for glycolysis and tumor development in esophageal squamous cell carcinoma (ESCC). Mechanistically, they show that *AGPG* can interact with the glycolytic enzyme PFKFB3 and somehow this interaction is important to maintain PFKFB3 protein stability.

In recent years many lncRNAs have been identified to regulate glycolysis or cancer metabolism through diverse array of mechanisms. This study identified one more such lincRNA. The novelty and broad impact of the study is considered to be relatively moderate. In addition, there are several significant weaknesses that prevent its publication in Nature Communications in its current format. The specific comments are:

1. Based on the data from Fig. 1, lncRNA *AGPG* is mainly localized in nucleus. There are wide concerns on using shRNA to study nucleus-localized lncRNAs, because shRNA mainly works for cytoplasm-localized RNA, such as mRNAs encoding proteins. The authors showed that they can rescue the phenotypes from shRNA studies by restoring the cDNA of this lncRNA. This approach is commonly used in studying protein-coding genes by restoring a cDNA with silent mutation on the corresponding sites in shRNA. However, it would be difficult to use this approach for lncRNA (or any non-coding gene). So it remains unclear how they can restore the expression of *AGPG* because overexpressed *AGPG* should still be targeted by its shRNA. The authors need to confirm their key findings with CRISPR technology, which has been commonly used in studying lncRNAs now.

Response: We apologize for not providing a clear explanation and evidence. In our study, *AGPG* was mainly localized in the nucleus, with some localization in the cytoplasm. Our previous results showed that shRNA was effective (Figure S2A); therefore, we performed functional experiments with shRNA, which has been used in some nuclear-localized lncRNA studies²⁶. However,

following the reviewer's suggestions, we also confirmed our findings with CRISPR/Cas9 technology. As shown in Figures 3J-3K and S3F-S3I, *AGPG* KO in ESCC cells strikingly inhibited glycolysis, proliferation and cell cycle progression, further confirming that *AGPG* is essential for ESCC glycolytic reprogramming, cell proliferation and cell cycle progression. Overexpression of *AGPG* FL, but not *AGPG* Δ T5, was sufficient to prevent the phenotypes observed after *AGPG* KO, including glycolytic reprogramming, cell proliferation and cell cycle regulation (Figures 3J-3K, S3G-S3I, Page 9, Line 168). These results suggest that the T5 fragment is required for *AGPG* function in interacting with and regulating PFKFB3.

We also confirmed that *AGPG* is required for PFKFB3 stabilization in the CRISPR/Cas9-based experiments. *AGPG* KO markedly reduced PFKFB3 expression, which was rescued by overexpression of *AGPG* FL but not of *AGPG* Δ T5 (Figure 4D, Page 9, Line 181). Our previous results showed that *AGPG* stabilizes PFKFB3 by preventing K302 ubiquitination. Using CRISPR technology, we also confirmed that the regulatory roles of *AGPG* in cell proliferation and glycolytic reprogramming are dependent on PFKFB3. PFKFB3 K302A overexpression significantly reversed the inhibition of glycolysis, cell proliferation and cell cycle progression by *AGPG* KO, while PFKFB3 WT only partially rescued these effects (Figures 4K-4M, S4K-S4L, Page 11, Line 213).

2. Another major weakness of this study is the lack of detailed mechanism by which *AGPG* regulates PFKFB3 protein stability. It is hard to image that, without any other protein partner, this lncRNA is sufficient to stabilize a protein. Does *AGPG* promote PFKFB3 protein stability through regulating another E3 ubiquitin ligase or DUB? The authors need to present more mechanistic data to address this question.

Response: In our study, we performed a variety of *in vitro* and *in vivo* experiments to study the interaction between the lncRNA and the protein, including RNA pull-down, RNA immunoprecipitation (RIP), CLIP (an improved method for the isolation of lncRNA segments bound to PFKFB3, Figure 3H), and MTRAP (*AGPG* lncRNA was modified to include an MS2 binding site and expressed in cells to show that PFKFB3 was associated with *AGPG* lncRNA *in vivo*, Figure 3D). Consistently, we demonstrated that *AGPG* directly associated with PFKFB3, which implies that PFKFB3 is a newly discovered noncanonical RBP (Page 15, Line 292). These findings underscore the complexity of posttranscriptional regulation of cellular systems.

In our study, *AGPG* KO markedly reduced PFKFB3 expression, which was rescued by overexpression of *AGPG* FL (Figure 4D). The effects might be attributable to proteasomal degradation, as decreased PFKFB3 levels were recovered by the proteasomal inhibitor MG-132 (Figure 4E). In addition, *AGPG* knockdown had marked effects on PFKFB3 stabilization in ESCC cells, significantly shortening the half-life of PFKFB3 (Figures 4F, S4B-S4C). Moreover, we carried out IP assays in cells expressing FLAG-tagged PFKFB3; proteins were immunoprecipitated with an anti-FLAG antibody, and ubiquitin levels were detected by western blotting. *AGPG* knockdown significantly increased the ubiquitination of PFKFB3 (Figures 4G, S4D). As suggested in many previous reports^{24, 28, 29}, these data suggest that *AGPG* is required for PFKFB3 stabilization because it prevents PFKFB3 ubiquitination.

As reported previously, PFKFB3 contains a KEN box, which targets proteins for ubiquitylation by APC/C⁶. It has been reported that PFKFB3 is subject to degradation involving APC/C-Cdh1, a cell cycle-regulated E3 ubiquitin ligase. Therefore, to further elucidate the mechanism by which *AGPG* blocks PFKFB3 ubiquitination, we performed coIP assays to identify whether *AGPG* affects the interaction between PFKFB3 and APC/C. Active APC/C was immunoprecipitated from ESCC cells using an anti-Cdc27 antibody (Abcam,

ab10538). As shown in Figure 4H, *AGPG* KO significantly increased the interaction of PFKFB3 and Cdc27, suggesting that *AGPG* blocks the binding of Cdc27 to PFKFB3. Collectively, the data indicate that *AGPG* specifically binds to PFKFB3 and blocks its interaction with APC/C, which inhibits APC/C-mediated PFKFB3 ubiquitination and subsequent degradation (Page 10, Line 195).

3. The study showed that *AGPG* deficiency resulted in defects in glycolysis, cell proliferation, and tumor growth, with decreased PFKFB3 levels. However, whether *AGPG* regulates these cellular functions THROUGH PFKFB3 remains unclear. It is still possible that *AGPG* regulates cancer metabolism through other targets. The authors need to examine whether restoring PFKFB3 K302A (a mutant that is resistant to *AGPG* regulation) in *AGPG* deficient cells would at least partially rescue defects in glycolysis, cell proliferation, and tumor development caused by *AGPG* deficiency.

Response: Thank you for making this good point. Following the reviewer's suggestion, we performed a series of rescue experiments to investigate the role of PFKFB3 in *AGPG*-mediated cell proliferation and glycolytic reprogramming. PFKFB3 K302A overexpression significantly reversed the inhibition of glycolysis, cell proliferation and cell cycle progression by *AGPG* KO, while PFKFB3 WT only partially rescued these effects (Figures 4K-4M, S4K-S4L, Page 11, Line 213). We also ascertained the effect of *AGPG* on PFKFB3 KO cell lines. After PFKFB3 KO, *AGPG* KO had mild effects on aerobic glycolysis, proliferation and cell cycle regulation in ESCC cells (Figure S4M-S4O, Page 11, Line 218). Collectively, these data suggest that the regulatory roles of *AGPG* in cell proliferation and glycolytic reprogramming are mainly dependent on PFKFB3.

4. The study used delta T5 of *AGPG* as a non-binding mutant to PFKFB3. However, there is no data to show that delta T5 is indeed a non-binding mutant. Fig. 3G and 3H showed that T1, T2, T3, or T4 does not interact with PFKFB3, but they never examine the binding between T1+2+3+4 (which is delta T5) and PFKFB3.

Response: Thank you for this suggestion. We apologize for not providing data showing that *AGPG* Δ T5 is indeed a nonbinding mutant. We performed RNA pull-down assays (revised Figure 3I), and after deleting the T5 fragment, *AGPG* could no longer interact with PFKFB3, which suggests that the T5 fragment is required for the *AGPG*-PFKFB3 interaction (Page 9, Line 168).

5. For patient survival analyses, it is unclear why some analyses choose 50%-50% to separate low vs high expression (which is the standard way for such analyses), whereas other data (such as Fig. 1E) choose other ways for separation. All analyses should be conducted using 50%-50% separation.

Response: Thank you for the suggestion. In our revised manuscript, all survival analyses were conducted using a 50%-50% separation as suggested. In the revised Figure 1E, we used this method to separate ESCC patients into groups with low or high *AGPG* expression, and consistent with our previous results, high *AGPG* levels correlated with an unfavorable overall survival (TCGA, n = 161).

6. In Fig. 6F, they delivered *AGPG* siRNA *in vivo* to PDX models, and showed this impacted on tumor development. However, there is no data to document that siRNA indeed decreased *AGPG* expression in tumors. They need to show this.

Response: Thank you for the suggestion. We apologize for not providing data showing that *in vivo*-optimized RNA interference indeed decreased *AGPG*

expression in tumors. And the main component of the *in vivo*-optimized *AGPG* inhibitor used in the PDX model was actually Antisense Oligonucleotides (ASO) (Page 13, Line 257), we apologize for not providing an accurate description before. Also, we detected the *AGPG* expression in PDX tumor tissues by q-PCR analysis, as shown in Figure S6G, *in vivo* optimized *AGPG* inhibitor significantly inhibited the *AGPG* expression in the tumors.

Minor points:

1. Fig. 1G shows that, while *AGPG* is overexpressed in some cancers, in many other cancers there is no difference or its level is even significantly decreased in several cancers, such as KIRP and KIRC. The authors need to comment on this.

Response: Thank you for the suggestion. We apologize for not comprehensively explaining *AGPG* expression levels in different cancers. In our revised manuscript (Page 6, Line 108), we have included the following text: “However, in some cancers, such as glioblastoma multiforme (GBM), head and neck squamous cell carcinoma (HNSC) and thyroid carcinoma (THCA), there was no significant difference between tumor and normal tissues. In addition, *AGPG* levels were decreased in cancers such as kidney chromophobe (KICH), kidney renal clear cell carcinoma (KIRC), and kidney renal papillary cell carcinoma (KIRP). Similar to many other lncRNAs, *AGPG* has tissue-specific expression patterns in different cancers³⁰.”

2. The study identified PFKFB3 as a *AGPG*-interacting protein based on MS analysis. They need to include the list of the proteins (including peptide numbers) identified from MS.

Response: Thank you for the suggestion. We apologize for not providing the

RNA pull-down mass spectrometry raw data. The mass spectrometry data including peptide numbers are provided in supplementary Table S7 (Page 7, Line 140). PFKFB3 was identified as an *AGPG*-associated protein.

3. Page 14 states that “However, very few lncRNAs have been implicated in cancer metabolism regulation, and the underlying mechanisms remain poorly understood.” This clearly is an overstatement. In recent years, many lncRNAs have been identified to regulate cancer metabolism. There are now many reviews to discuss and summarize this research topic. At the best, this study identified the first lncRNA to directly regulate PFKFB3.

Response: Thank you for noting this issue. We apologize for our one-sided comments on this issue. We have modified these comments in our revised manuscript (Page 3, Line 58) as follows: “In recent years, many lncRNAs have been identified to regulate cancer metabolism, but the underlying mechanisms remain elusive. Here, we identified for the first time that the lncRNA actin gamma 1 pseudogene (*AGPG*) plays a pivotal role in glucose metabolism remodeling and cell proliferation by enhancing PFKFB3 stability. Intriguingly, this is the first lncRNA shown to directly bind to and regulate PFKFB3.”

Reference

1. Hafner, M. *et al.* Transcriptome-wide Identification of RNA-Binding Protein and MicroRNA Target Sites by PAR-CLIP. *CELL* **141**, 129-141 (2010).
2. Engreitz, J.M. *et al.* Local regulation of gene expression by lncRNA promoters, transcription and splicing. *NATURE* **539**, 452-455 (2016).
3. Cabili, M.N. *et al.* Localization and abundance analysis of human lncRNAs at single-cell and single-molecule resolution. *GENOME BIOL* **16**, 20 (2015).
4. Hon, C.C. *et al.* An atlas of human long non-coding RNAs with accurate 5' ends.

NATURE **543**, 199-204 (2017).

5. Liu, S. *et al.* Identification of lncRNA MEG3 Binding Protein Using MS2-Tagged RNA Affinity Purification and Mass Spectrometry. *APPL BIOCHEM BIOTECH* **176**, 1834-1845 (2015).

6. Herrero-Mendez, A. *et al.* The bioenergetic and antioxidant status of neurons is controlled by continuous degradation of a key glycolytic enzyme by APC/C–Cdh1. *NAT CELL BIOL* **11**, 747-752 (2009).

7. Ullah, S. *et al.* dbPAF: an integrative database of protein phosphorylation in animals and fungi. *Sci Rep* **6**, 23534 (2016).

8. Liu, Z. *et al.* CPLM: a database of protein lysine modifications. *NUCLEIC ACIDS RES* **42**, D531-D536 (2014).

9. Liu, W. *et al.* IBS: an illustrator for the presentation and visualization of biological sequences. *BIOINFORMATICS* **31**, 3359-3361 (2015).

10. Kim, W. *et al.* Systematic and quantitative assessment of the ubiquitin-modified proteome. *MOL CELL* **44**, 325-340 (2011).

11. Xu, G., Paige, J.S. & Jaffrey, S.R. Global analysis of lysine ubiquitination by ubiquitin remnant immunoaffinity profiling. *NAT BIOTECHNOL* **28**, 868-873 (2010).

12. Wagner, S.A. *et al.* A proteome-wide, quantitative survey of in vivo ubiquitylation sites reveals widespread regulatory roles. *MOL CELL PROTEOMICS* **10**, M111-M13284 (2011).

13. Oshikawa, K., Matsumoto, M., Oyamada, K. & Nakayama, K.I. Proteome-wide identification of ubiquitylation sites by conjugation of engineered lysine-less ubiquitin. *J PROTEOME RES* **11**, 796-807 (2012).

14. Li, F. *et al.* Acetylation accumulates PFKFB3 in cytoplasm to promote glycolysis and protects cells from cisplatin-induced apoptosis. *NAT COMMUN* **9** (2018).

15. Ramanathan, M., Porter, D.F. & Khavari, P.A. Methods to study RNA–protein interactions. *NAT METHODS* **16**, 225-234 (2019).

16. Shi, L., Pan, H., Liu, Z., Xie, J. & Han, W. Roles of PFKFB3 in cancer. *Signal Transduct Target Ther* **2**, 17044 (2017).

17. Doménech, E. *et al.* AMPK and PFKFB3 mediate glycolysis and survival in response to mitophagy during mitotic arrest. *NAT CELL BIOL* **17**, 1304-1316 (2015).

18. Marsin, A., Bouzin, C., Bertrand, L. & Hue, L. The Stimulation of Glycolysis by Hypoxia in Activated Monocytes Is Mediated by AMP-activated Protein Kinase and Inducible 6-Phosphofructo-2-kinase. *J BIOL CHEM* **277**, 30778-30783 (2002).

19. Wang, Y. *et al.* CPT1A-mediated fatty acid oxidation promotes colorectal cancer cell

metastasis by inhibiting anoikis. *ONCOGENE* (2018).

20. Horikoshi, N. *et al.* Two domains of p53 interact with the TATA-binding protein, and the adenovirus 13S E1A protein disrupts the association, relieving p53-mediated transcriptional repression. *MOL CELL BIOL* **15**, 227-234 (1995).

21. Hong, T.M., Chen, J.J., Peck, K., Yang, P.C. & Wu, C.W. p53 amino acids 339-346 represent the minimal p53 repression domain. *J BIOL CHEM* **276**, 1510-1515 (2001).

22. Godar, S. *et al.* Growth-inhibitory and tumor-suppressive functions of p53 depend on its repression of CD44 expression. *CELL* **134**, 62-73 (2008).

23. Miki, T., Matsumoto, T., Zhao, Z. & Lee, C.C. p53 regulates Period2 expression and the circadian clock. *NAT COMMUN* **4**, 2444 (2013).

24. Hu, W.L. *et al.* GUARDIN is a p53-responsive long non-coding RNA that is essential for genomic stability. *NAT CELL BIOL* **20**, 492-502 (2018).

25. Phatarpekar, P.V., Lee, D.A. & Somanchi, S.S. Electroporation of siRNA to Silence Gene Expression in Primary NK Cells. *Methods Mol Biol* **1441**, 267-276 (2016).

26. Li, W. *et al.* Increased Levels of the Long Intergenic Non-Protein Coding RNA POU3F3 Promote DNA Methylation in Esophageal Squamous Cell Carcinoma Cells. *GASTROENTEROLOGY* **146**, 1714-1726 (2014).

27. Yu, X.X. *et al.* Antisense oligonucleotide reduction of DGAT2 expression improves hepatic steatosis and hyperlipidemia in obese mice. *HEPATOLOGY* **42**, 362-371 (2005).

28. Jiang, R. *et al.* The long noncoding RNA lnc-EGFR stimulates T-regulatory cells differentiation thus promoting hepatocellular carcinoma immune evasion. *NAT COMMUN* **8**, 15129 (2017).

29. Zheng, J. *et al.* Pancreatic cancer risk variant in LINC00673 creates a miR-1231 binding site and interferes with PTPN11 degradation. *NAT GENET* **48**, 747-757 (2016).

30. Mattioli, K. *et al.* High-throughput functional analysis of lncRNA core promoters elucidates rules governing tissue specificity. *GENOME RES* **29**, 344-355 (2019).

Reviewers' comments:

Reviewer #1 (Remarks to the Author):

In the revised version of the manuscript, the authors have completed a significant amount of work to perform additional experiments. The new data have significantly strengthened the functional role of lncRNA AGPG, providing further support for their model. However, there are still critical concerns, listed below, that should be addressed before this study is given further consideration.

Major Points:

1) As mentioned in the previous review, PFKFB3 CLIP experiment was required to identify the short nucleotide motifs of AGPG responsible for PFKFB3 binding. Although authors performed CLIP-qPCR assay, the results do not show any specific motifs responsible for PFKFB3 binding. Furthermore, if PFKFB3 is a bona fide RNA-binding protein, it should bind more RNAs in cell than just one single RNA AGPG. Only the classic CLIP assay would answer the above questions. This must be tested, however, and not solely speculated, to be the case.

2) Regarding the raw data in supplementary table 7, AGPG pull-down mass spectrometry identified over hundreds of AGPG-binding proteins but I had a hard time to follow author's rationale to select PFKFB3 for further study. What are the criteria that authors used to exclude the other proteins in the list? Furthermore, beads only and AGPG antisense pull-down mass spectrometry data are missing, raising the concern of unspecific binding. If the authors do not address this point, their model would be possibly misleading.

Reviewer #2 (Remarks to the Author):

The authors conducted additional experiments to address the previous concerns from this reviewer. However, the new data also raised additional significant concerns (see below for detailed description). Overall, the study presented a large amount of data without paying sufficient attention to clarity and consistency.

In response to this reviewer's previous request, the authors claimed that they generated AGPG KO cells using CRISPR. However, such information is not clear from their data or text. It seems that the earliest place where they introduced AGPG CRISPR KO cells is at lines 168-171 (Overexpression of AGPG FL, but not a mutant lacking the T5 fragment (AGPG Δ T5), was sufficient to prevent the phenotypes observed after AGPG knockout (KO), including glycolytic reprogramming, cell proliferation and cell cycle regulation (Figures 3J-3K, S3F-S3I).). However, before this point, the manuscript never introduced or described how they generated AGPG CRISPR KO cells and what are the phenotypes from AGPG CRISPR KO cells. They should have described this in Fig. 2 (in which they described the phenotypes of AGPG knockdown cells by shRNA). Following their shRNA data in Fig.2, they need to state that to further confirm their data from shRNA knockdown, they also generated CRISPR KO cells, followed by the description of the phenotypes in AGPG CRISPR KO cells.

Along the similar line, how they generated AGPG CRISPR KO cells was never clearly described in the manuscript (including the method). KO a lncRNA using CRISPR is not a trivial task. Therefore, they need to describe this in detail. In text, the authors need to briefly describe how they generate the cells with a schematic to show the gRNA design in AGPG genomic locus. In the method they need to provide gRNA sequence information and more experimental details on exactly how they generated CRISPR KO cells (for example, how they screened lncRNA CRISPR KO cells). Without such information, it is impossible for other researchers to repeat their findings in the future.

When they say AGPG KO, it is often unclear whether this refers to AGPG CRISPR KO or AGPG

knockdown cells by shRNA. For example, lines 182-183 stated "AGPG KO markedly reduced PFKFB3 expression, which was rescued by overexpression of AGPG FL but not AGPG Δ T5 183 (Figure 4C-4D)". However, the corresponding western blotting data in Figure 4C shows AGPG knockdown, followed by Fig. 4D again using the term AGPG KO (CRISPR KO or shRNA knockdown)? There are multiple such examples throughout the manuscript, and the reviewer cannot list all of them. To be clarified, in each figure throughout their manuscript the authors need to clearly state whether they used CRISPR KO or shRNA KD cells.

Figure S4G showed that AGPG KO does not affect PFKFB3 S461 phosphorylation. (Again, AGPG KO here means CRISPR KO or shRNA KD? The data before here used shRNA.) Before this data, the authors just showed that "AGPG knockdown had marked effects on PFKFB3 stabilization in ESCC cells, shortening the half-life of PFKFB3 (Figures 4F, S4B-S4C)." If AGPG knockdown or KO does not affect PFKFB3 S461 phosphorylation, AGPG deficiency should still increase its phosphorylation because PFKFB3 protein level is markedly increased upon AGPG deficiency (according to their data in Fig. 4C). How do the authors explain this discrepancy?

Lines 197-200 states "Therefore, to further elucidate the mechanism by which AGPG blocks PFKFB3 ubiquitination, we performed coIP assays to determine whether AGPG affects the interaction between PFKFB3 and APC/C. Notably, AGPG KO significantly increased the PFKFB3/Cdc27 interaction (Figure 4H), suggesting that AGPG could block the binding of Cdc27 to PFKFB3." In this one sentence, they shifted from the term APC/C to Cdc27, which is very confusing. In addition, Fig. 4H needs to show whole cell lysates as loading control for PFKFB3 and APC/C under all experimental conditions.

Fig. S4I showed that AGPG stabilizes PFKFB3 through regulating its K302 ubiquitination. Since the new data showed that AGPG regulates PFKFB3 ubiquitination through controlling APC/C interaction with PFKFB3, this data would indicate that APC/C regulates K302 ubiquitination in PFKFB3. The authors need to confirm this (or if the previous publication (ref 22) already identified this as the major site for APC/C-mediated ubiquitination in PFKFB3, they need to point it out).

Fig. S4M-4O is designed to address whether the effect of AGPG KO on glycolysis etc is mediated through PFKFB3 (by examining the effect of AGPG KO in PFKFB3 KO setting). However, this experiment lacks PFKFB3 WT setting as an important control. They need to present the data in four settings: control, AGPG KO, PFKFB3 KO, AGPG+PFKFB3 double KO.

Fig. S1H shows three bar graphs comparing AGPG expression between normal (N) and tumor (T) samples in ESCC, GC, and CRC. It is unclear which bar graph corresponds to which tumor type (the information was not provided in either figure or figure legend). They need to label the tumor type in each bar graph.

Lines 153-156 (page 8) stated "Absolute quantitation of AGPG and PFKFB3 levels showed that there were ~400–700 AGPG molecules per cell versus ~4400–7400 PFKFB3 molecules per cell (Figure S3D), indicating that there are sufficient AGPG copies in ESCC cells and that the endogenous AGPG-PFKFB3 interaction might occur frequently." Fig. S3D measured RNA copy numbers of AGPG and PFKFB3, but AGPG (RNA) interacts with PFKFB3 (protein). It is unclear to this reviewer how the measurement of PFKFB3 RNA copy numbers could lead to the authors to conclude the interaction of AGPG with PFKFB3 protein occurs frequently.

Line 46-47 states that "Long noncoding RNAs (lncRNAs) are suggested to be involved in metabolic reprogramming, but the mechanisms remain elusive" and cites reference 4. However, reference 4 is a research article (rather than a review) which has little to do with lincRNA function in cancer metabolism.

Dear reviewers,

We would like to express our utmost gratitude to you for taking the time and effort to review our manuscript and provide constructive advice and suggestions for our paper. We have taken this advice into consideration and made the appropriate changes; hopefully, the improvements are acceptable. All changes are identified by page and line location and noted by highlighting or strikethrough in the revised manuscript with tracked changes. Our detailed responses are as follows.

Reviewers' comments:

Reviewer #1 (Remarks to the Author):

In the revised version of the manuscript, the authors have completed a significant amount of work to perform additional experiments. The new data have significantly strengthened the functional role of lncRNA AGPG, providing further support for their model. However, there are still critical concerns, listed below, that should be addressed before this study is given further consideration.

Major Points:

1) As mentioned in the previous review, PFKFB3 CLIP experiment was required to identify the short nucleotide motifs of AGPG responsible for PFKFB3 binding. Although authors performed CLIP-qPCR assay, the results do not show any specific motifs responsible for PFKFB3 binding. Furthermore, if PFKFB3 is a bona fide RNA-binding protein, it should bind more RNAs in cell than just one single RNA AGPG. Only the classic CLIP assay would answer the above questions. This must be tested, however, and not solely speculated, to be the case.

Response: Thank you for the suggestion. As you suggested, we performed crosslinking-immunoprecipitation and high-throughput sequencing (HITS-CLIP). Hypergeometric Optimization of Motif EnRichment (Homer) was used for motif analysis based on the binding peaks obtained by the Piranha and CIMS methods^{1, 2}. The experimental details of HITS-CLIP are described in the Methods section. The RNA motifs recognized by PFKFB3 are listed in Table S5. Among these motifs, CCAGCCA or similar motifs were highly ranked and could be identified by multiple methods. Therefore, CCAGCCA in the T5 fragment of AGPG might mediate PFKFB3

binding. (Page 9, Line 180)

2) Regarding the raw data in supplementary table 7, AGPG pull-down mass spectrometry identified over hundreds of AGPG-binding proteins but I had a hard time to follow author's rationale to select PFKFB3 for further study. What are the criteria that authors used to exclude the other proteins in the list? Furthermore, beads only and AGPG antisense pull-down mass spectrometry data are missing, raising the concern of unspecific binding. If the authors do not address this point, their model would be possibly misleading.

Response: Thank you for noting this issue. The antisense AGPG pull-down mass spectrometry data are provided in Table S9. A list of the antisense AGPG binding proteins from the MS analysis is also provided. In our revised manuscript, we explained why we selected PFKFB3 for further study, as follows: "We compared the AGPG binding proteins with antisense AGPG binding proteins. Proteins that bound to antisense AGPG were excluded from the candidate list, and the remaining proteins were sorted by MS score, as described in previous lncRNA studies³." (Page 8, Line 143) Thus, PFKFB3 ranked first among the remaining proteins. In addition, the AGPG-PFKFB3 interaction was further confirmed by the finding that AGPG bound directly to purified His-tagged recombinant PFKFB3 (Figure 3B), as well as the RNA immunoprecipitation (RIP) assays (Figures 3C, S3A-S3B). Therefore, we believe that AGPG is associated with PFKFB3.

Reviewer #2 (Remarks to the Author):

The authors conducted additional experiments to address the previous concerns from this reviewer. However, the new data also raised additional significant concerns (see below for detailed description). Overall, the study presented a large amount of data without paying sufficient attention to clarity and consistency.

In response to this reviewer's previous request, the authors claimed that they generated AGPG KO cells using CRISPR. However, such information is not clear from their data or text. It seems that the earliest place where they introduced AGPG CRISPR KO cells is at lines 168-171 (Overexpression of AGPG FL, but not a mutant lacking the T5 fragment (AGPG Δ T5), was sufficient to prevent the phenotypes observed after AGPG knockout (KO), including glycolytic reprogramming, cell

proliferation and cell cycle regulation (Figures 3J-3K, S3F-S3I).). However, before this point, the manuscript never introduced or described how they generated AGPG CRISPR KO cells and what are the phenotypes from AGPG CRISPR KO cells. They should have described this in Fig. 2 (in which they described the phenotypes of AGPG knockdown cells by shRNA). Following their shRNA data in Fig.2, they need to state that to further confirm their data from shRNA knockdown, they also generated CRISPR KO cells, followed by the description of the phenotypes in AGPG CRISPR KO cells.

Along the similar line, how they generated AGPG CRISPR KO cells was never clearly described in the manuscript (including the method). KO a lncRNA using CRISPR is not a trivial task. Therefore, they need to describe this in detail. In text, the authors need to briefly describe how they generate the cells with a schematic to show the gRNA design in AGPG genomic locus. In the method they need to provide gRNA sequence information and more experimental details on exactly how they generated CRISPR KO cells (for example, how they screened lncRNA CRISPR KO cells). Without such information, it is impossible for other researchers to repeat their findings in the future.

Response: Thank you for noting this issue. We apologize for not clearly describing how we generated the AGPG CRISPR KO cells before. In our revised manuscript, we have added this information as follows (Page 19, Line 379): “AGPG KO cell lines were generated using a CRISPR/Cas9-based strategy⁴. AGPG-specific guide RNA (gRNA) expression vectors were obtained from Kidan Biotechnology Co., Ltd. (Guangzhou, China). Briefly, AGPG-specific sgRNAs were designed to recognize two different sites of the AGPG gene, the location of the upstream target (>hg38_refGene_NR_002929_0 range=chr1:202861697-202861719) and the location of the downstream target (>hg38_refGene_NR_002929_6 range=chr1:202876356-202876378). To identify the effective sgRNAs, we transfected different sgRNA-Cas9 vectors into 293H cells, and then, the genomic DNA was extracted, Q5 PCR was performed and the PCR products were subjected to sequencing. The results showed that AGPG sgRNA01 (CGGCGGGGCTGTTTCGTAAG) was effective among the upstream sgRNAs, and AGPG sgRNA02 (ATCAAGTGTCTATATGCGT) was effective among the downstream sgRNAs. The PCR primer GCAACACCACGAATCCCAAC/TTGTCCCGCTCTGGAAACTC and the sequencing primer TTGTCCCGCTCTGGAAACTC were used to detect the efficiency of AGPG sgRNA01. The PCR primer AACTAGGCCATGCACCAA/GCCCACAGGCCAAATTCATTC and the sequencing primer GCCTCAGCCCACAGAGCTTA were used to detect the efficiency of AGPG sgRNA02. Thus, double sgRNA vectors were constructed using the sgRNAs mentioned above. After the

construction of the vectors, the vectors were sequenced and compared with the target genes, which showed that the vectors were constructed correctly. ESCC cells were transfected with the pLV-U6-AGPG RNA sgRNA01-7SK-sgRNA02-EFS-hCas9-2A-Puro or pLV-U6-NC sgRNA01-7SK-NC sgRNA02-EFS-hCas9-2A-Puro expression vectors. The transfected cells were selected using puromycin (2 µg/ml) for seven days. Isolated single colonies were expanded and subjected to detection of genomic deletions by PCR.”

As you suggested, we added the phenotypes of the AGPG CRISPR KO cells in Figure S2E-S2I and described these data as follows: “To further confirm the functional role of AGPG, we also generated AGPG CRISPR KO cells using the CRISPR/Cas9 genome editing system (Figure S2E). Consistently, AGPG CRISPR KO significantly inhibited ESCC cell proliferation and cell cycle progression (Figure S2F-S2G). Additionally, AGPG CRISPR KO led to a significant reduction in aerobic glycolysis (Figure S2H-S2I). Taken together, our data indicated that AGPG is functionally important in regulating cancer metabolic reprogramming and tumor growth.” (Page 7, Line 136)

When they say AGPG KO, it is often unclear whether this refers to AGPG CRISPR KO or AGPG knockdown cells by shRNA. For example, lines 182-183 stated “AGPG KO markedly reduced PFKFB3 expression, which was rescued by overexpression of AGPG FL but not AGPG Δ T5 183 (Figure 4C-4D)”. However, the corresponding western blotting data in Figure 4C shows AGPG knockdown, followed by Fig. 4D again using the term AGPG KO (CRISPR KO or shRNA knockdown)? There are multiple such examples throughout the manuscript, and the reviewer cannot list all of them. To be clarified, in each figure throughout their manuscript the authors need to clearly state whether they used CRISPR KO or shRNA KD cells.

Response: Thank you for this suggestion. In our revised manuscript, we added "AGPG CRISPR KO" to all figures and texts when we used AGPG CRISPR KO cells. We used “sh AGPG” in all figures when we used shRNA KD cells. We apologize for not clearly describing the data you mentioned. In Figure 4C-4D, we corrected the description as follows: “AGPG knockdown mediated by shRNA significantly reduced the expression of PFKFB3 (Figure 4C), which was also confirmed in AGPG CRISPR KO cells using CRISPR/Cas9 (Figure 4D).” (Page 10, Line 193)

Figure S4G showed that AGPG KO does not affect PFKFB3 S461 phosphorylation. (Again, AGPG KO here means CRISPR KO or shRNA KD? The data before here used shRNA.) Before this data,

the authors just showed that “AGPG knockdown had marked effects on PFKFB3 stabilization in ESCC cells, shortening the half-life of PFKFB3 (Figures 4F, S4B-S4C).” If AGPG knockdown or KO does not affect PFKFB3 S461 phosphorylation, AGPG deficiency should still increase its phosphorylation because PFKFB3 protein level is markedly increased upon AGPG deficiency (according to their data in Fig. 4C). How do the authors explain this discrepancy?

Response: Thank you for noting this issue. In Figure S4G, we have added "AGPG CRISPR KO" to state that we used AGPG CRISPR KO cells here. We apologize for not providing a clear description in this section. Because AGPG CRISPR KO led to a significant decrease in PFKFB3 protein levels, we treated the cells with the proteasomal inhibitor MG-132 to inhibit PFKFB3 degradation so that we could sufficiently enrich S461 phosphorylated PFKFB3 for western blotting and compare the PFKFB3 S461 phosphorylation levels, this method was also used in other studies⁵. The PFKFB3 total protein levels were also provided in Figure S4G. Therefore, we concluded that AGPG CRISPR KO has little effect on PFKFB3 S461 phosphorylation.

Lines 197-200 states “Therefore, to further elucidate the mechanism by which AGPG blocks PFKFB3 ubiquitination, we performed coIP assays to determine whether AGPG affects the interaction between PFKFB3 and APC/C. Notably, AGPG KO significantly increased the PFKFB3/Cdc27 interaction (Figure 4H), suggesting that AGPG could block the binding of Cdc27 to PFKFB3.” In this one sentence, they shifted from the term APC/C to Cdc27, which is very confusing. In addition, Fig. 4H needs to show whole cell lysates as loading control for PFKFB3 and APC/C under all experimental conditions.

Response: Thank you for noting this issue. In our revised manuscript, we explained Figure 4H as follows (Page 11, Line 211): “PFKFB3 was shown to be subject to degradation involving APC/C-Cdh1, a cell cycle-regulated E3 ubiquitin ligase. APC/C is composed of multiple subunits; as reported previously, active APC/C could be immunoprecipitated from cells using a monoclonal Cdc27 antibody^{6, 7}. Therefore, to further elucidate the mechanism by which AGPG blocks PFKFB3 ubiquitination, we performed coIP assays using PFKFB3 and Cdc27 antibodies to determine whether AGPG affects the interaction between PFKFB3 and active APC/C. AGPG CRISPR KO significantly increased the PFKFB3/Cdc27 interaction (Figure 4H), suggesting that AGPG could block the binding of Cdc27 to PFKFB3. Therefore, we speculate that AGPG specifically binds to PFKFB3 and blocks its interaction with APC/C, which inhibits APC/C-mediated PFKFB3

ubiquitination and the subsequent degradation.” As you suggested, we have provided whole cell lysates as loading controls for PFKFB3 and APC/C under all experimental conditions in Figure 4H.

Fig. S4I showed that AGPG stabilizes PFKFB3 through regulating its K302 ubiquitination. Since the new data showed that AGPG regulates PFKFB3 ubiquitination through controlling APC/C interaction with PFKFB3, this data would indicate that APC/C regulates K302 ubiquitination in PFKFB3. The authors need to confirm this (or if the previous publication (ref 22) already identified this as the major site for APC/C-mediated ubiquitination in PFKFB3, they need to point it out).

Response: Thank you for noting this issue. In our revised manuscript, we further confirmed that APC/C regulates PFKFB3 K302 ubiquitination in ESCC cells, which is consistent with our data and previous studies⁷. Since Cdc27 is a key component of APC/C⁶, we overexpressed Cdc27 in ESCC cells to verify whether APC/C induces PFKFB3 ubiquitination. In Figures S4J-S4K, PFKFB3 K302A, but not other mutants, significantly abrogated APC/C (Cdc27)-induced PFKFB3 ubiquitination in ESCC cells, further indicating that K302 is an important site for APC/C-mediated ubiquitination in PFKFB3. (Page 11, Line 225)

Fig. S4M-4O is designed to address whether the effect of AGPG KO on glycolysis etc is mediated through PFKFB3 (by examining the effect of AGPG KO in PFKFB3 KO setting). However, this experiment lacks PFKFB3 WT setting as an important control. They need to present the data in four settings: control, AGPG KO, PFKFB3 KO, AGPG+PFKFB3 double KO.

Response: Thank you for this suggestion. In our revised manuscript, we designed four groups—Ctrl, AGPG CRISPR KO, PFKFB3 CRISPR KO, and AGPG CRISPR KO+PFKFB3 CRISPR KO—to further confirm that the effect of AGPG is mainly mediated through PFKFB3. Consistent with our previous data, after PFKFB3 KO in ESCC cells, AGPG CRISPR KO had mild effects on aerobic glycolysis, cell proliferation and cell cycle regulation (Figure S5A-S5C).

Fig. S1H shows three bar graphs comparing AGPG expression between normal (N) and tumor (T) samples in ESCC, GC, and CRC. It is unclear which bar graph corresponds to which tumor type (the information was not provided in either figure or figure legend). They need to label the tumor type in each bar graph.

Response: Thank you for noting this issue. In Fig. S1H, the bar graph corresponds to ESCC, GC, and CRC in Figure 1I. The tumor type has been labeled in each bar graph of Fig. S1H.

Lines 153-156 (page 8) stated “Absolute quantitation of AGPG and PFKFB3 levels showed that there were ~400–700 AGPG molecules per cell versus ~4400–7400 PFKFB3 molecules per cell (Figure S3D), indicating that there are sufficient AGPG copies in ESCC cells and that the endogenous AGPG-PFKFB3 interaction might occur frequently.” Fig. S3D measured RNA copy numbers of AGPG and PFKFB3, but AGPG (RNA) interacts with PFKFB3 (protein). It is unclear to this reviewer how the measurement of PFKFB3 RNA copy numbers could lead to the authors to conclude the interaction of AGPG with PFKFB3 protein occurs frequently.

Response: Thank you for noting this issue. According to our data and previous studies^{5,8}, the copy number of AGPG is not low, indicating that there are sufficient copies of AGPG in ESCC cells to form the AGPG-PFKFB3 interaction. However, we did not verify that “the interaction of AGPG with PFKFB3 protein occurs frequently”, so we apologize for this unverified hypothesis. To prevent confusion, we have removed this inappropriate statement in our revised manuscript.

Line 46-47 states that “Long noncoding RNAs (lncRNAs) are suggested to be involved in metabolic reprogramming, but the mechanisms remain elusive” and cites reference 4. However, reference 4 is a research article (rather than a review) which has little to do with lincRNA function in cancer metabolism.

Response: Thank you for noting this issue. We apologize for this inappropriate citation. Reference 4 has been removed in our revised manuscript. To further illustrate the advances in lncRNA and metabolic regulation, we cited two other references related to this issue^{9,10}. (Page 3, Line 47)

Once again, we hope that these responses are adequate and would like to express our gratitude to you for your time and considerations in reviewing our manuscript.

Best Regards,

Dr. Xu

Reference:

1. Uren, P.J. et al. Site identification in high-throughput RNA-protein interaction data. *BIOINFORMATICS* 28, 3013-3020 (2012).
2. Zhang, C. & Darnell, R.B. Mapping in vivo protein-RNA interactions at single-nucleotide resolution from HITS-CLIP data. *NAT BIOTECHNOL* 29, 607-614 (2011).

3. Xing, Z. et al. Expression of Long Noncoding RNA YIYA Promotes Glycolysis in Breast Cancer. *CANCER RES* 78, 4524-4532 (2018).
4. Zhuo, W. et al. Long Noncoding RNA GMAN, Up-regulated in Gastric Cancer Tissues, Is Associated With Metastasis in Patients and Promotes Translation of Ephrin A1 by Competitively Binding GMAN-AS. *GASTROENTEROLOGY* 156, 676-691 (2019).
5. Hu, W.L. et al. GUARDIN is a p53-responsive long non-coding RNA that is essential for genomic stability. *NAT CELL BIOL* 20, 492-502 (2018).
6. Cedeno, C., La Monaca, E., Esposito, M. & Gutierrez, G.J. Detection and Analysis of Cell Cycle-Associated APC/C-Mediated Cellular Ubiquitylation In Vitro and In Vivo. *Methods Mol Biol* 1449, 251-265 (2016).
7. Herrero-Mendez, A. et al. The bioenergetic and antioxidant status of neurons is controlled by continuous degradation of a key glycolytic enzyme by APC/C–Cdh1. *NAT CELL BIOL* 11, 747-752 (2009).
8. Cabili, M.N. et al. Localization and abundance analysis of human lncRNAs at single-cell and single-molecule resolution. *GENOME BIOL* 16, 20 (2015).
9. Chen, F. et al. Extracellular vesicle-packaged HIF-1alpha-stabilizing lncRNA from tumour-associated macrophages regulates aerobic glycolysis of breast cancer cells. *NAT CELL BIOL* 21, 498-510 (2019).
10. Shankaraiah, R.C., Veronese, A., Sabbioni, S. & Negrini, M. Non-coding RNAs in the reprogramming of glucose metabolism in cancer. *CANCER LETT* 419, 167-174 (2018).

Reviewers' comments:

Reviewer #1 (Remarks to the Author):

There are flaws in PFKFB3 CLIP experiment, and missing of several key data has been the major problems that make it wholly unconvincing. So, although authors' conclusion seems like an interesting idea, and the effects of AGPG knockdown on PFKFB3-mediated tumor glycolytic reprogramming are certainly intriguing, the data to support the proposed mechanism are just not convincing at this stage.

Specific points:

1. The PFKFB3 CLIP experiment does not seem to have the key autoradiography gel showing the endogenous PFKFB3-RNA complex in cell as well as a western blot control showing that the PFKFB3 immunoprecipitation is specific. Given that this assay is also performed in the absence of PFKFB3 knockdown or knockout condition, the possibility that the motif shown in Figure 3L is a non-specific artifact is not excluded – and so the reported PFKFB3- AGPG may not be reliable.

2. The methods section describing the CLIP approach does not provide any information on the adapter sequence used. An accession number for raw and processed CLIP data is not provided. It is impossible to infer how many independent CLIP experiments were conducted and whether replicates are reproducible.

3. The authors claimed that CCAGCCA or similar motifs were highly ranked PFKFB3 binding motifs and one of such motif in the T5 fragment of AGPG might mediate PFKFB3 binding. Even if this is so, the evidence as to whether this sequence actually is the PFKFB3 binding region must be presented. Most importantly, if this study is to be taken seriously in the field, the nucleotides of AGPG that coordinate the PFKFB3 binding must be identified. It should be possible to generate mutants of this lncRNA and test them for their ability to bind PFKFB3 and for their ability to promote tumor glycolytic reprogramming in a rescue experiment.

4. Table S5 is completely missing.

Reviewer #2 (Remarks to the Author):

The authors have adequately addressed the remaining concerns from this reviewer. The paper can be accepted for publication.

Dear reviewer,

We would like to express our utmost gratitude to you for taking the time and effort to review our manuscript and provide such detailed comments on our paper. We also thank you for the positive comments about our manuscript. We apologize for not providing sufficient data regarding the CLIP-seq analysis. Here, we have taken into account all the points raised; hopefully, the improvements are acceptable. All changes are identified by the page and line location and noted by highlighting or strikethrough in the revised manuscript with tracked changes. Our detailed responses are as follows.

Reviewers' comments:

Reviewer #1 (Remarks to the Author):

There are flaws in PFKFB3 CLIP experiment, and missing of several key data has been the major problems that make it wholly unconvincing. So, although authors' conclusion seems like an interesting idea, and the effects of *AGPG* knockdown on PFKFB3-mediated tumor glycolytic reprogramming are certainly intriguing, the data to support the proposed mechanism are just not convincing at this stage.

Specific points:

1. The PFKFB3 CLIP experiment does not seem to have the key autoradiography gel showing the endogenous PFKFB3-RNA complex in cell as well as a western blot control showing that the PFKFB3 immunoprecipitation is specific. Given that this assay is also performed in the absence of PFKFB3 knockdown or knockout condition, the possibility that the motif shown in Figure 3L is a non-specific artifact is not excluded – and so the reported PFKFB3- *AGPG* may not be reliable.

Response: Thank you for the suggestion. For this part, we used an alternative method for the CLIP protocol because the use of radioisotopes is limited in China¹. References were added to our revised manuscript (Page 31, Line 665, Reference 57). Briefly, RBP-RNA interactions are stabilized with UV crosslinking, followed by limited RNase I digestion, immunoprecipitation of the RBP-RNA complexes with a specific antibody for the protein of interest, and stringent washes. After dephosphorylation of RNA fragments, an “in-line-barcoded” RNA adapter is ligated to the 3' end.

After protein gel electrophoresis and nitrocellulose membrane transfer, a region 75 kDa (~220 nt of RNA) above the protein size is excised, and proteinase K is added to isolate the RNA. RNA is further prepared for paired-end high-throughput sequencing libraries. The experimental details for CLIP are described in the Methods section (Page 23, Line 458). As you suggested, the raw and processed CLIP data are also provided in our revised manuscript. In our study, we performed a variety of *in vitro* and *in vivo* experiments to study the interaction between *AGPG* and PFKFB3, including RNA pull-down assays, RNA immunoprecipitation, MS2-tagged RNA affinity purification and CLIP. We consistently demonstrated that *AGPG* is associated with PFKFB3.

2.The methods section describing the CLIP approach does not provide any information on the adapter sequence used. An accession number for raw and processed CLIP data is not provided. It is impossible to infer how many independent CLIP experiments were conducted and whether replicates are reproducible.

Response: Thank you for the suggestion. We apology for not providing these data before. As you suggested, we have added the adapter sequence to our revised manuscript. The raw and processed CLIP data were uploaded to NCBI Sequence Read Archive, and the accession number was provided. We have added this information as follows: “The End1 3' adapter 5'-AGATCGGAAGAGC-3' and the End2 3' adaptor 5'-AGATCGGAAGAGC-3' were used in this study. The CLIP-Seq dataset is available at NCBI Sequence Read Archive (SRA) under BioProject PRJNA591321”. (Page 24, Line 479)

3.The authors claimed that CCAGCCA or similar motifs were highly ranked PFKFB3 binding motifs and one of such motif in the T5 fragment of *AGPG* might mediate PFKFB3 binding. Even if this is so, the evidence as to whether this sequence actually is the PFKFB3 binding region must be presented. Most importantly, if this study is to be taken seriously in the field, the nucleotides of *AGPG* that coordinate the PFKFB3 binding must be identified. It should be possible to generate mutants of this lncRNA and test them for their ability to bind PFKFB3 and for their ability to promote tumor glycolytic reprogramming in a rescue experiment.

Response: Thank you for the suggestion. In our previous data, we demonstrated that after deleting the T5 fragment, *AGPG* could no longer interact with PFKFB3 (Figure 3I). Overexpression of *AGPG* FL, but not a mutant lacking the T5 fragment (*AGPG* Δ T5), was sufficient to prevent the

phenotypes observed after *AGPG* knockout (KO), including glycolytic reprogramming (Figures 3J-3K, S3G-S3I). These results suggest that the T5 fragment is required for *AGPG* to interact with and regulate PFKFB3. Consistently, our CLIP data suggested that the CCAGCCA motif in the T5 fragment might mediate PFKFB3 binding (Figure 3L). As you suggested, to further verify whether this motif coordinates PFKFB3 binding, we performed RNA pull-down assays using wild type *AGPG* (WT) and CCAGCCA motif-deleted *AGPG* (MT). We demonstrated that the binding of a mutant lacking the CCAGCCA motif (*AGPG* MT) and PFKFB3 was significantly reduced (Figure S3J). In addition, overexpression of this *AGPG* MT could not rescue the decreased glycolysis caused by *AGPG* KO (Figure S3K). These data suggest that the CCAGCCA motif of *AGPG* is important for its ability to bind PFKFB3 and promote tumor glycolytic reprogramming. (Page 10, Line 185)

4. Table S5 is completely missing.

Response: Table S5 is in the file “Supplementary information”. (Page 9)

We hope that these responses are adequate and would like to express our gratitude to you for your time and considerations in reviewing our manuscript.

Best Regards,

Dr. Xu

Reference:

1. Van Nostrand, E.L. *et al.* Robust transcriptome-wide discovery of RNA-binding protein binding sites with enhanced CLIP (eCLIP). *NAT METHODS* **13**, 508-514 (2016).

Reviewers' comments:

Reviewer #1 (Remarks to the Author):

The authors have addressed some of my remaining concerns. However, without the key autoradiography gel showing the endogenous PFKFB3-RNA complex in cell as well as a western blot control showing that the PFKFB3 immunoprecipitation is specific, the possibility that PFKFB3-AGPG interaction is an artifact can not be excluded. Furthermore, it is not clear how authors were able to cut the RNA-protein complex bands without autoradiography gel as the reference. Also, did authors cut the same band in IgG CLIP sample for sequencing to show that CCAGCCA motif is not enriched?

Reviewer #3 (Remarks to the Author):

This reviewer only comments on PFKFB3 binding specificity based on in vitro binding experiments and CLIP data. Overall, the in vitro data showing the specific binding of PFKFB3 binding to AGPG depending on the T5 fragments containing the identified CCAGCCA motif appears to be convincing. The comparison of WT and mutant in cells is also supportive.

The CLIP experiments were performed using a modified protocol. Specifically, crosslinked protein-RNA complexes were cut from SDS page directly without membrane transfer (a step to remove free non-crosslinked RNA and other background). This is not ideal, and it may increase background. This is not to suggest the motif identified in the study unreliable, but with the limited information provided, it is difficult to make a clear judgement. The minimum the authors could provide is the distribution of CLIP tags on AGPG together with IgG control. An unambiguous peak overlapping with the motif will support the claim of the authors.

Dear reviewers,

We would like to express our utmost gratitude to you for taking the time and effort to review our manuscript and provide such detailed comments on our paper. We also thank you for the positive comments about our manuscript. Here, we have taken into account all the points raised; hopefully, the improvements are acceptable. All changes are identified by the page and line location and noted by highlighting or strikethrough in the revised manuscript with tracked changes. Our detailed responses are as follows.

Reviewers' comments:

Reviewer #1 (Remarks to the Author):

The authors have addressed some of my remaining concerns. However, without the key autoradiography gel showing the endogenous PFKFB3-RNA complex in cell as well as a western blot control showing that the PFKFB3 immunoprecipitation is specific, the possibility that PFKFB3-AGPG interaction is an artifact can not be excluded. Furthermore, it is not clear how authors were able to cut the RNA-protein complex bands without autoradiography gel as the reference. Also, did authors cut the same band in IgG CLIP sample for sequencing to show that CCAGCCA motif is not enriched?

Response: Thank you for the suggestion. PFKFB3 immunoprecipitation has been proved to be specific in our RIP assays, where an IgG control was used (Figure S3B). The same antibody was used in our CLIP experiment. As shown below, we have confirmed the specificity of PFKFB3 immunoprecipitation again.

Fig.1 The western blot showed that the PFKFB3 immunoprecipitation is specific

The CLIP experiments were performed using a modified protocol. After protein gel electrophoresis, the band of the protein of interest and the region 75 kDa above it were cut from the gel according to the molecular weight of the protein. The figures of the relevant experiments are shown below.

Fig.2 The band of the protein of interest and the region 75 kDa above it were cut from the gel.

Fig.3 The protein marker used in the CLIP experiment.

Furthermore, our previous experiments such as RNA pulldown assays with purified recombinant PFKFB3, RNA immunoprecipitation assays (with IgG control), MS2-tagged RNA affinity purification (MTRAP), cross-linking immunoprecipitation and qPCR (CLIP-qPCR) have more than shown enough evidence that PFKFB3 binds specifically to *AGPG*.

In the previously reported CLIP-seq¹⁻⁴, the input can also serve as a crucial control for nonspecific background signal, so we did not carry out an IgG CLIP sample for sequencing. Of course, we agree that the control group is essential

in the study. Therefore, various controls have been used in our previous experiments to verify the binding specificity of PFKFB3 and *AGPG*, such as IgG control in RIP assays, beads only and antisense control in RNA pulldown assays, pcDNA3.1-MS2 control in MS2-tagged RNA affinity purification, and so on. By comparing the input and IP data, we found CCAGCCA or similar motifs were highly ranked and could be identified by multiple methods. To further verify whether this motif coordinates PFKFB3 binding, we performed RNA pull-down assays using wild type *AGPG* (WT) and CCAGCCA motif-deleted *AGPG* (MT). We demonstrated that the binding of a mutant lacking the CCAGCCA motif (*AGPG* MT) and PFKFB3 was significantly reduced (Figure S3J). In addition, overexpression of this *AGPG* MT could not rescue the decreased glycolysis caused by *AGPG* KO (Figure S3K). These data suggest that the CCAGCCA motif of *AGPG* is important for its ability to bind PFKFB3 and promote tumor glycolytic reprogramming.

We sincerely hope the reviewer finds the evidence obtained from our above mentioned experiment proof enough that PFKFB3 binds specifically to *AGPG*.

Reviewer #3 (Remarks to the Author):

This reviewer only comments on PFKFB3 binding specificity based on in vitro binding experiments and CLIP data. Overall, the in vitro data showing the specific binding of PFKFB3 binding to AGPG depending on the T5 fragments containing the identified CCAGCCA motif appears to be convincing. The comparison of WT and mutant in cells is also supportive.

The CLIP experiments were performed using a modified protocol. Specifically, crosslinked protein-RNA complexes were cut from SDS page directly without membrane transfer (a step to remove free non-crosslinked RNA and other background). This is not ideal, and it may increase background. This is not to suggest the motif identified in the study unreliable, but with the limited information provided, it is difficult to make a clear judgement. The minimum the authors could provide is the distribution of CLIP tags on AGPG together with IgG control. An unambiguous peak overlapping with the motif will support the claim of the authors.

Response: Thank you for the positive comments and the helpful suggestions. The main concern about our last submission is the binding specificity of PFKFB3 and AGPG. Actually, our previous experiments such as RNA pulldown assays with purified recombinant PFKFB3, RNA immunoprecipitation assays (with IgG control), MS2-tagged RNA affinity purification (MTRAP), cross-linking immunoprecipitation and qPCR (CLIP-qPCR) have shown enough evidence that PFKFB3 binds specifically to AGPG.

Our previous data have suggested that the T5 fragment is required for AGPG to interact with and regulate PFKFB3. According to Reviewer #1's suggestion, we further performed CLIP-seq to identify the motif of AGPG that are responsible for PFKFB3 binding. We demonstrated that CCAGCCA or similar motifs were highly ranked and could be identified by multiple methods during the CLIP-seq analysis. To further verify whether this motif coordinates PFKFB3 binding, we performed RNA pull-down assays and other rescue experiments using wild type AGPG (WT) and CCAGCCA motif-deleted AGPG (MT). Consistently, those data suggest that CCAGCCA motif on the T5 fragment is

important for its ability to bind PFKFB3 and promote tumor glycolytic reprogramming.

We deeply appreciate you for recognizing our data and claims. As you suggested, we have re-analyzed the CLIP-seq data and as expected, we did find that the PFKFB3 CLIP sequencing reads mapping to *AGPG* covered the previously identified motif, which further implied that the PFKFB3 can bind directly to *AGPG*. As shown below, compared with the input, PFKFB3 CLIP enriched more reads mapping to the *AGPG* sequence around the previously identified motif. However, the number of reads seems not to be ideal, which may be caused by the low abundance of lncRNAs and the insufficient depth of sequencing^{2, 5}, but this does not affect the general logic behind the binding between PFKFB3 and *AGPG*. Our CLIP-seq data was mainly used to further identify the key region for PFKFB3-*AGPG* interaction, and the current CLIP-seq results combined with the results from previous experiments all point to PFKFB3 binding specifically to *AGPG* at least partially through the identified motif.

Fig 1. The CLIP sequencing reads mapping to *AGPG* were visualized by IGV

In the previously reported CLIP-seq¹⁻⁴, the input can also serve as a crucial

control for nonspecific background signal, so we did not carry out an IgG CLIP sample for sequencing. Of course, we agree that the control group is essential in a study. Therefore, various controls have been used in our previous experiments to verify the binding specificity of PFKFB3 and AGPG, such as IgG control in RIP assays, beads only and antisense control in RNA pulldown assays, pcDNA3.1-MS2 control in MS2-tagged RNA affinity purification, and so on. Nevertheless, if the reviewer still feels that the IgG control is indispensable to the rigorousness of our CLIP-seq, we could manage repeating this experiment, however it will incur an unnecessary waste of time and experimental supplies.

We hope that these responses are adequate and would like to express our gratitude to you for your time and considerations in reviewing our manuscript.

Best Regards,

Dr. Xu & Dr. Ju

Reference:

1. Van Nostrand, E.L. *et al.* Robust transcriptome-wide discovery of RNA-binding protein binding sites with enhanced CLIP (eCLIP). *NAT METHODS* **13**, 508-514 (2016).
2. Lin, C. & Miles, W.O. Beyond CLIP: advances and opportunities to measure RBP–RNA and RNA–RNA interactions. *NUCLEIC ACIDS RES* **47**, 5490-5501 (2019).
3. Takeuchi, A. *et al.* Loss of Sfpq Causes Long-Gene Transcriptopathy in the Brain. *CELL REP* **23**, 1326-1341 (2018).
4. Zhang, X., Chen, X., Liu, Q., Zhang, S. & Hu, W. Translation repression via modulation of the cytoplasmic poly(A)-binding protein in the inflammatory response. *ELIFE* **6** (2017).
5. Cabili, M.N. *et al.* Localization and abundance analysis of human lncRNAs at single-cell and single-molecule resolution. *GENOME BIOL* **16**, 20 (2015).

Reviewer #3 (Remarks to the Author):

I have no more comments on the manuscript.

Reviewers' comments:

Reviewer #3 (Remarks to the Author):

I have no more comments on the manuscript.

Response: We would like to express our utmost gratitude to you for taking the time and effort to review our manuscript. Thank you very much for the positive comments and the helpful suggestions.

Best Regards,

Dr. Xu & Dr. Ju